# PDSketch: Integrated Planning Domain Programming and Learning

**Jiayuan Mao**[1]   **Tomás Lozano-Pérez**[1]   **Joshua B. Tenenbaum**[1,2,3]   **Leslie Pack Kaelbling**[1]

[1] MIT Computer Science & Artificial Intelligence Laboratory
[2] MIT Department of Brain and Cognitive Sciences
[3] Center for Brains, Minds and Machines

## Abstract

This paper studies a model learning and online planning approach towards building flexible and general robots. Specifically, we investigate how to exploit the *locality* and *sparsity* structures in the underlying environmental transition model to improve model generalization, data-efficiency, and runtime-efficiency. We present a new domain definition language, named PDSketch. It allows users to flexibly define high-level structures in the transition models, such as object and feature dependencies, in a way similar to how programmers use TensorFlow or PyTorch to specify kernel sizes and hidden dimensions of a convolutional neural network. The details of the transition model will be filled in by trainable neural networks. Based on the defined structures and learned parameters, PDSketch automatically generates domain-independent planning heuristics without additional training. The derived heuristics accelerate the performance-time planning for novel goals.

## 1   Introduction

A long-standing goal in AI is to build robotic agents that are flexible and general, able to accomplish a diverse set of tasks in novel and complex environments. Such tasks generally require a robot to generate long-horizon plans for novel goals in novel situations, by reasoning about many objects and the ways in which their state-changes depend on one another. A promising solution strategy is to combine model learning with online planning: the agent forms an internal representation of the environment's states and dynamics by learning from external or actively-collected data, and then applies planning algorithms to generate actions, given a new situation and goal at performance time.

There are two primary desiderata for a system based on model-learning and planning. First, the learning process should be *data efficient*, especially because of the combinatorial complexity of possible configuration of in the real world. Second, the learned model should be *computationally efficient*, making online planning a feasible runtime execution strategy.

A critical strategy for learning models that generalize well from small amounts of data and that can be deployed efficiently at runtime is to introduce inductive biases. In image processing, we leverage translation invariance and equivariance by using convolutions. In graph learning, we leverage permutation invariance by using graph neural networks. Two essential forms of structure we can leverage in dynamic models of the physical world are *locality* and *sparsity*. Consider a robot picking an object up off a table. At the object level, the operation only changes the states of the object being picked up and the robot, and only depends on a few other nearby objects, such as the table (local). At the object feature level, only the configuration of the robot and pose of the object are changed, but their colors and frictional properties are unaffected (sparse).

Classical hand-engineered approaches to robot task and motion planning have designed representations that expose and exploit locality through *lifting* (or *object-centrism*), which allows relational

---

Correspondence to: jiayuanm@mit.edu. Project page: http://pdsketch.csail.mit.edu.

36th Conference on Neural Information Processing Systems (NeurIPS 2022).

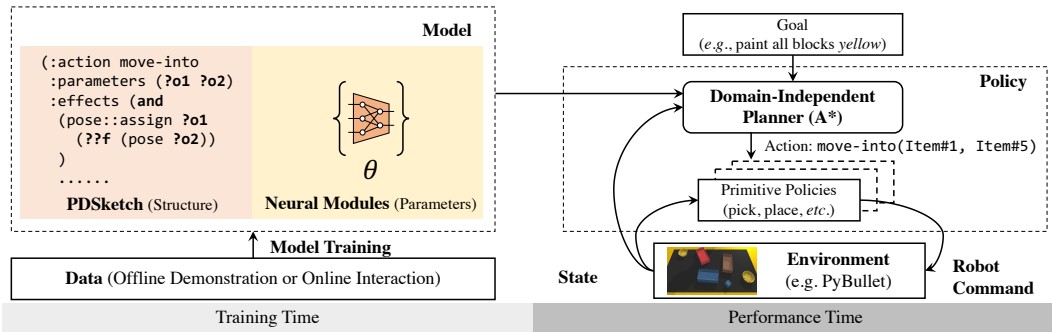

Figure 1: The life cycle of a PDSketch model. A PDSketch model is composed of a model structure definition and a collection of trainable neural modules. The model parameters can be learned from data. During performance time, the model is used by a domain-independent planner to form a policy that directly interacts with the environment.

descriptions of objects and abstraction over them, and through *factoring*, which represents different attributes of an object in a disentangled way [Garrett et al., 2021]. These representations are powerful and effective articulations of locality and sparsity, but they are traditionally laboriously hand-designed in a process that is very difficult to get correct, similar to writing

```
def move_into(o, c):
  o.pose = c.pose + [0, 0, 0.1]
  if is_block(o) and is_painter(c):
    o.color = c.color
```

(a) A full specification of the transition model.

```
def move_into(o, c):
  o.prop1 = ??(c.prop1)
  if ??(o, c):
    o.prop2 = ??(c.prop2)
```

(b) A structure-only specification.

Figure 2: Defining both the transition model structure and implementation in Python (a) vs. defining only the structure while leaving details (the **??** functions) to be learned (b).

a full state-transition function as in Fig. 2a. This approach is not directly applicable to problems involving perception or environmental dynamics that are unknown or difficult to specify. In this paper, we present PDSketch, a model-specification language that integrates human specification of structural sparsity priors and machine learning of continuous and symbolic aspects of the model. Just as human users may define the structure of a convolutional neural network in TensorFlow [Abadi et al., 2016] or PyTorch [Paszke et al., 2019], PDSketch allows users to specify *high-level* structures of the transition model as in Fig. 2b (analogous to setting the kernel sizes), and uses machine learning to fill in the details (analogous to learning the convolution kernels).

Fig. 1 depicts the life-cycle of a PDSketch model, PDSketch uses an object-centric, factored, symbolic language to flexibly describe structural inductive biases in planning domains (i.e., the model structure). A PDSketch model is associated with a collection of neural modules whose parameters can be learned from robot trajectory data that are either collected offline by experts or actively-collected by interacting with the environment. During performance time, a PDSketch is paired with a domain-independent planner, such as $A^*$, and as a whole forms a goal-conditioned policy. The planner receives the environmental state and the trained PDSketch model, makes plans in an abstract action space, and invokes primitive policies that actually generate robot joint commands.

Compared to unstructured models, such as a single multi-layer perceptron that models the complete state transition monolithically, the structures specified in PDSketch substantially improve model generalization and data-efficiency in training. In addition, they enable the computation of powerful *domain-independent planning heuristics*: these are estimates of the cost-to-go from each state to a state satisfying the goal specification, which can be obtained from the structured transition model *without any additional learning*. They can be leveraged by $A^*$ to efficiently plan for *unseen* goals, specified in a first-order logic language.

We experimentally verify the efficiency and effectiveness of PDSketch in two domains: BabyAI, an 2D grid-world environment that focuses on navigation, and Painting Factory, a simulated table-top robotic environment that paints and moves blocks. Our results suggest that 1) locality and sparsity structures, specified economically in a few lines of code, can significantly improve the data efficiency of model learning; 2) the model learning and planning paradigm enables strong generalization to unseen goal specifications. Finally, the domain-independent heuristics automatically induced from the structures dramatically improve performance-time efficiency, especially for novel goal specifications.

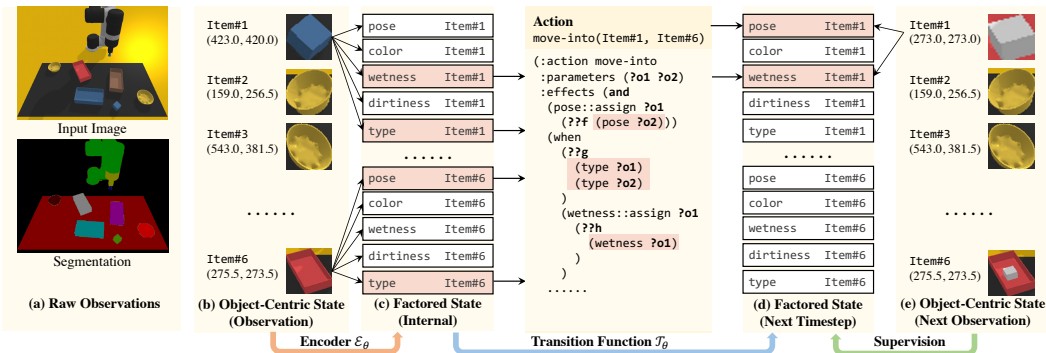

Figure 3: A factorized state representation and transition model for the robot painting domain. The raw observation (a) is first processed by an external perception module into an object-centric representation (b). This representation is further transformed into a fine-grained factorization (c). The transition function $\mathcal{T}_\theta$ can be defined over this factored state representation: each action may only change a few factors of the state. Executing a specific action `move-into(item#1, item#6)` produces the predicted factored state at the next timestep. During training time, we will be using the object-centric observation from the next timestep to supervise the learning of $\mathcal{T}_\theta$.

## 2    PDSketch

We focus on the problem of learning models for a robot that operates in a space $\mathcal{S}$ of world states that plans to achieve goal conditions that are subsets of $\mathcal{S}$. A planning problem is a tuple $\langle \mathcal{S}, s_0, g, \mathcal{A}, \mathcal{T} \rangle$, where $s_0 \in \mathcal{S}$ is the initial state, $g$ is a goal specification in first-order logic, $\mathcal{A}$ is a set of actions that the agent can execute, and $\mathcal{T}$ is a environmental transition model $\mathcal{T}: \mathcal{S} \times \mathcal{A} \to \mathcal{S}$. The task of planning is to output a sequence of actions $\bar{a} = \{a_i \in \mathcal{A}\}$ in which the terminal state $s_T$ induced by applying $a_i$ sequentially following $\mathcal{T}$ satisfies the goal specification $g$: $eval(g, s_T) = 1$. The function $eval(g, s)$ determines whether state $s$ satisfies the goal condition $g$ by recursively evaluating the logical expression and using learned neural groundings of the primitive terms in the expression.

At execution time, the agent will observe $s_0$ and be given $g$ from human input, such as a first-order logic expression corresponding to "all the apples are in a blue bowl." However, we do not assume that the agent knows, in advance, the groundings of $g$ (i.e. the underlying $eval(g, s)$ function) or the transition model $\mathcal{T}$. Thus, we need to learn $g$ and $\mathcal{T}$ from data, in the form of observed trajectories that achieve goal states of interest.

Formally, we assume the training data given to the agent is a collection of tuples $\langle \bar{s}, \bar{a}, g, \overline{succ} \rangle$, where $\bar{s} = \langle s_i \rangle$ is a sequence of world states, $\bar{a} = \langle a_i \rangle$ is the sequence of actions taken by the robot, $g$ is a goal specification, and $\overline{succ} = \langle succ_i \rangle$ is the "task-success" signal. Each $succ_i \in \{0, 1\}$ indicates whether the goal $g$ is satisfied at state $s_i$: $succ_i = eval(g, s_i)$. The data sequences should be representative of the dynamics of the domain but need not be optimal goal-reaching trajectories.

It can be difficult to learn a transition model that is accurate over the long term on some types of state representations. For this reason, we generally assume an arbitrary latent space, $\Phi$, for planning. The learning problem, then, is to find three parametric functions, collectively parameterized by $\theta$: state encoder $\mathcal{E}_\theta: \mathcal{S} \to \Phi$, goal-evaluation function $eval_\theta: \mathcal{G} \times \Phi \to \{0, 1\}$, and transition model $\mathcal{T}_\theta: \Phi \times \mathcal{A} \to \Phi$. Although the domain might be mildly partially observable or stochastic, our goal will be to recover the most accurate possible deterministic model on the latent space.

**Local, sparse structure.** We need models that will generalize very broadly to scenarios with different numbers and types of objects in widely varying arrangements. To achieve this, we exploit structure to enable compositional generalization: throughout this work we will be committing to an object-centric representation for $s$, a logical language for goals $g$, and a sparse, local model of action effects.

We begin by factoring the environmental state $s$ into a set of object states. Each $s \in \mathcal{S}$ is a tuple $(\mathcal{U}_s, f_s)$, where $\mathcal{U}_s$ is the set of objects in state $s$, denoted by arbitrary unique names (e.g., `item#1`, `item#2`). The object set $\mathcal{U}_s$ is assumed to be constant within a single episode, but may differ in different episodes. The second component, $f_s$, is a dictionary mapping each object name to a fixed-dimensional object-state representation, such as a local image crop of the object and its position. We

can extend this representation to relations among objects, for example by adding $g_s(x, y), x, y \in \mathcal{U}$ as a mapping from each object pair to a vector representation. We assume the detection and tracking of objects through time is done by external perception modules (e.g., object detectors and trackers).

We carry the object-centric representation through to actions, goals, and the transition model. Specifically, we define a *predicate* as a tuple $\langle name, args, grounding \rangle$, where *args* is a list of $k$ arguments and *grounding* is a function from the latent representations of the objects corresponding to its arguments (in $\Psi^k$) into a scalar or vector value. (This is a generalization of the typical use of the term "predicate," which is better suited for use in robotics domains in which many quantities we must reason about are continuous.) For example, as illustrated in Fig. 3c, the predicate `wetness` takes a single argument as its input, and returns a (latent) vector representation of its wetness property; its grounding might be a neural network that maps from the visual appearance of the object to the latent wetness value. Given a set of predicates, we define the language of possible goal specifications to be all first-order logic formulas over the subset of the predicates whose output type is Boolean. To evaluate a goal specification in a state $(\mathcal{U}, f)$, quantification is interpreted in the finite domain $\mathcal{U}$ and $f_s$ provides an interpretation of object names into representations that can serve as input to the grounded predicates.

The transition model $\mathcal{T}_\theta$ is specified in terms of a set object-parameterized action schema $\langle name, args, precond, effect, \pi \rangle$, where *name* is a symbol, *args* is a list of symbols, *precond* and *effect* are descriptions of the action's effects, described in section 2.1, and $\pi$ is a parameterized primitive policy for carrying out the action in terms of raw perception and motor commands. These local policies can be learned via demonstration or reinforcement learning in a phase prior to the model-learning phase, constructed using principles of control theory, or a combination of these methods. The set of concrete actions $\mathcal{A}$ available in a state $s$ is formed by instantiating the action scheme with objects in universe $\mathcal{U}_s$. We assume the transition dynamics of the domain (i.e., the *effect* of each action schema) are well characterized in terms of the changes of properties and relations of objects and that the transition model is lifted in the sense that it can be applied to domain instances with different numbers and types of objects. In addition, we assume the dynamics are local and sparse, in the sense that effects of any individual action depend on and change only a small number attributes and relations of a few objects, and that by default all other objects and attributes are unaffected. Taking again action schema `move-into` as an example, shown in Fig. 3d, only the states of object #1 and #6 are relevant to this action (but not #2, #3, etc.), and furthermore, the action only changes the `pose` and `wetness` properties of `item#1` (but not the color and the type).

The factored representation also introduces a factored learning problem: instead of learning a monolithic neural network for $\mathcal{T}_\theta$ and $eval_\theta$, the problem is factored into learning the grounding of individual predicates that appear in goal formulas, as well as the transition function for individual factors that were changed by an action.

## 2.1   Representation Language

The overall specification of $eval_\theta$ and $\mathcal{T}_\theta$ can be decomposed into two parts: 1) the locality and sparsity structures and 2) the actual model parameters, $\theta$, such as neural network weights. We provide a symbolic language for human programmers to specify the locality and sparsity structure of the domain and methods for representing and learning $\theta$. If the human provides no structure, the model falls back to a plain object-centric dynamics model Zhu et al. [2018]. However, we will show that explicit encoding of locality and sparsity structures can substantially improve the data efficiency of learning and the computational efficiency of planning with the resulting models.

PDSketch is an extension of the planning-domain definition language [Fikes and Nilsson, 1971, Fox and Long, 2003], a widely used formalism that focuses exposing locality and sparsity structure in symbolic planning domains. The key extensions are 1) allowing vector values in the computation graph and 2) enabling the programmer to use "blanks", which are unspecified functions that will be filled in with neural networks learned from data. Thus, rather than specifying the model in full detail, the programmer provides only a "sketch" [Solar-Lezama, 2008]. The two key representational components of PDDL are predicates and action schemas (operators).

Fig. 4 shows a simple example of PDSketch definition of predicates. All three predicates take a single object `?o` of type `item` as their argument, and return either a floating-point vector or a scalar value from 0 to 1, indicating the score of a binary classifier. The `image` predicate simply refers to the raw image crop feature of the object. The `is-yellow` predicate's *grounding* takes a very simple form "`(??f (color ?o))`". The term `??f` defines a slot whose name is `f`. It takes only one argument, the

```
(:predicates ; input features
  (image [return_type=vector[float32], input] ?o - item)
)
(:derived
  (color [return_type=vector[float32]] ?o - item)
  (??f (image ?o))
)
(:derived
  (is-yellow ?o - item)   ; parameter and type
  (??f (color ?o))         ; function body
)
```

(a) A PDSketch definition of an input feature of each objects:
`(image ?o)`, and two derived feature/predicates:
`(color ?o)` and `(is-yellow ?o)`.

(b) The corresponding "computation graph" induced by the definitions.
`color::f` and `is-yellow::f` are customizable CNNs that are applied
identically to each object in the input state.

Figure 4: A minimal example of defining derived features and predicates with blanks "**??**".

`color` of the object `?o`, and outputs a classification score, which can be interpreted as the score of the object `?o` being yellow. The actual computation of the `yellow` predicate from the `color` value (as well as the computation of the `color` value from the `image` value) is instantiated in a neural network with trained parameters. The computation graph for the whole model can be built by recursively chaining the function bodies of predicate definitions.

Next, we illustrate how *locality* and *sparsity* structures can be specified for an action schema. Fig. 5 defines an action schema *name* `move-into` with two effect components. First, highlighted in blue, the action changes the pose of object `?o1` to a new pose that depends on the current pose of the second object `?o2`. Rather than hand-coding this detailed dependence, we leave the grounding blank. In addition, in our domain, the `wetness` of an object may be changed when the object is placed into a specific type of container. This is encoded by specifying a conditional effect using the keyword `when`, with two parts: 1) a Boolean-valued condition g of some other predicates on the state (in this case, the types of the two objects), and 2) the actual "effect", in this case, to change the `wetness` of `?o1` based on a function that considers the current `wetness` of `?o1`. The update will be applied only if the condition is true. To ensure the computation is differentiable, we make this condition "soft": let $w$ be the current wetness, $w'$ be the new wetness computed by function `??h`, and $c$ be the scalar condition value computed by function `??g`. The updated value of the wetness will be $cw' + (1-c)w$. Note that all the pose, wetness, and type representations can be arbitrary latent vectors computed by an encoder from the raw input. Thus, the "blanks" `??f`, `??g`, `??h` are indeed general neural networks. The effect definition here induces a corresponding computation graph of neural network weights and state representation tensors.

Like PDDL, PDSketch has full support of first-order logic, including Boolean operations (and, or, not) and finite-domain quantifiers ($\forall$ and $\exists$). They allow us to define more complex structures in the domain of interest. We present our full language and more examples in the supplementary material.

## 2.2 Model Learning and Planning with PDSketch

Let $\theta$ denote the collection of all learnable parameters required to complete a PDSketch domain definition into a full model. This includes the parameters of the state encoder, all predicate groundings, and the definitions of slot-update functions that were left blank in the sketch. Recall that our training data are tuples of three sequences and a goal formula $\langle \overline{s}, \overline{a}, g, \overline{succ} \rangle$. Fundamentally, our objective is to minimize a sum of two losses, one related to predicting the truth values of the goal formulas and one related to predicting the next state, given the previous state and action:

$$\mathcal{L}(\theta) = \sum_{\langle \overline{s}, \overline{a}, g, \overline{succ} \rangle \in \mathcal{D}} \left\{ \sum_i \text{BCE}\left(eval_\theta(g, \mathcal{E}_\theta(s_i)), succ_i\right) + \sum_i \text{L1}\left(\mathcal{T}_\theta(\mathcal{E}_\theta(s_i), a_i), \mathcal{E}_\theta(s_{i+1})\right) \right\},$$

where BCE is the binary cross-entropy classification loss and L1 is a regression loss. To avoid degenerate local optima, we add a "lookahead" loss term that combines both aspects, as detailed in the supplement. If the encoder $\mathcal{E}_\theta$ is constrained to be the identity, and there is no predicate-level structure, then the transition model essentially learns to be a per-object next-image predictor. More generally, including the encoder turns this into a bisimulation objective [Li et al., 2006]: we want to uncover a latent transition model that accurately predicts the reward signal (in this case, whether the goal is satisfied or not) but does not necessarily reconstruct the input state representation.

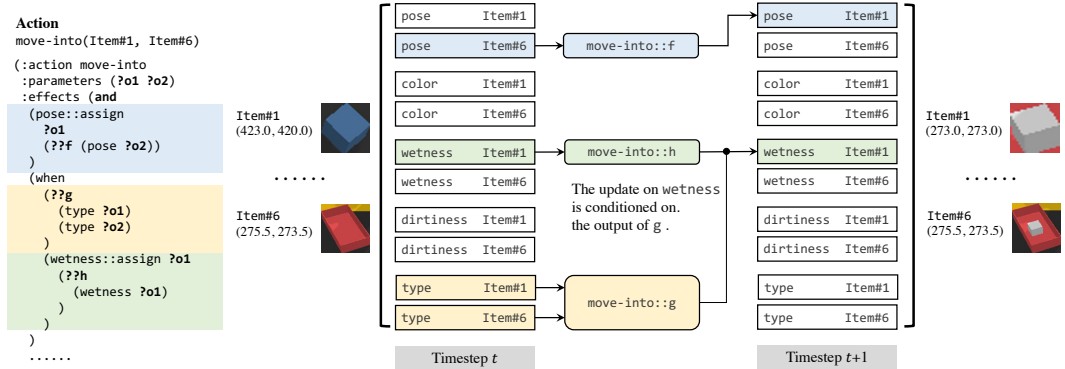

Figure 5: A computation graph of a partial definition of the action "move-into."

In order to adjust $\theta$ to minimize this loss, we must establish a differentiable computation graph. The encoder $\mathcal{E}_\theta$ will generally be a relatively standard combination of convolutional and fully-connected feed-forward neural network. Fig. 5 illustrates the computation graph associated with $\mathcal{T}_\theta$ for a particular choice of action $a_i$ and objects that serve as its arguments. Importantly, note that there is a substantial amount of parameter-tying: the same predicate-grounding network might appear in multiple times, even when characterizing $\mathcal{T}_\theta$ for a single action, if that predicate appears multiple times (applied to different objects) in the preconditions or effects of the action. The computation graph shown here, as well as those necessary to compute $eval_\theta(g, \mathcal{E}_\theta(s_i))$, involve Boolean operators, which do not have useful derivatives for optimization. We address this by representing truth values as elements of the interval $[0, 1]$ and approximate logical operations with the differentiable Gödel t-norms: $\texttt{not}(p) = 1 - p$, $\texttt{and}(p_1, p_2) = \min(p_1, p_2)$, $\forall x.p(x) = \min_x p(x)$.

Once we have estimated $\theta$ from data, we can solve any planning problem in the domain given any starting state $s_0 \in \mathcal{S}$ and goal $g$ expressed in terms of predicates for which we have groundings. The resulting transition model $\mathcal{T}$ can be used by a variety of different planners. We will focus on forward search, constructing a tree rooted at $\mathcal{E}(s_0)$ with branches corresponding to the possible instantiations, $a$, of action templates $\mathcal{A}$ with the objects in the universe $\mathcal{U}$ associated with $s_0$, and next latent states computed by applying $\mathcal{T}$. The search terminates when it reaches a node $n$ in which $eval_\theta(n, g) > .5$. Unguided forward search can be very slow when the planning horizon is long or the branching factor is large (e.g., when there are many objects in an environment). To address this, we will use the $A^*$ heuristic search algorithm using a *domain-independent search heuristic* directly derivable from the locality and sparsity structure defined in PDSketch.

### 2.3 Inducing Domain-Independent Heuristics

Because of the rich representational capacity of PDSketch models, in which values can be continuous and multidimensional, we cannot take advantage of the planning algorithms that operate on PDDL input, such as Fast Downward [Helmert, 2006], which derive much of their efficiency from domain-independent heuristics. We cannot use their strategies in detail, but we take inspiration from the idea of deriving an optimistic estimate of the cost to reach the goal from a state by solving a "relaxed" version of the problem, which is computationally easier than the original Bonet and Geffner [2001].

One way to construct a relaxed planning problem is to allow each predicate instance to take on multiple values at the same time. For example, the robot or an object can effectively be in multiple places at the same time. In this relaxation, computing the number of steps needed to change the value of a predicate instance can be done in polynomial time. We can use such a relaxation in the hFF heuristic [Hoffmann and Nebel, 2001] to get an estimate of the cost to goal, by first chaining the actions forward until all the components of the goal condition have been made true, and then searching to recover a small set of actions that can collectively achieve the goal under the relaxation.

To use hFF, however, we must reduce our continuous-space problem to a discrete-space one. Specifically, we discretizes all continuous state variables (poses, etc.) into a designated number of bins (e.g., 128). Next, for each externally-defined functions, we learn a first-order decision tree to approximate the computation (e.g., to approximate the neural network). Concretely, we use VQVAE [Van

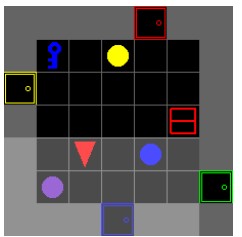

Figure 6: A screenshot of the BabyAI environment.

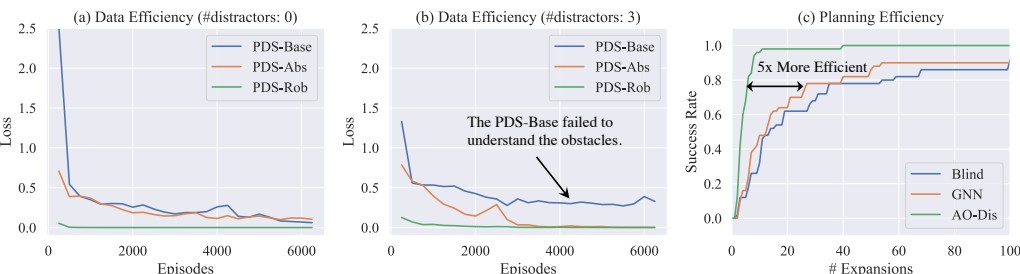

| Model | Inductive Biases | | | | Succ. Rate | |
|---|---|---|---|---|---|---|
| | Obj-Centric. | Facing | Rob. Dyn. | Obj. Prop. | Basic | #Obj.Gen. |
| BC | Y | N | N | N | 0.93 | 0.79 |
| DT (S) | Y | N | N | N | 0.91 | 0.82 |
| DT (S+F) | Y | N | N | N | 0.32 | 0.19 |
| DreamerV2 | N | N | N | N | 0.96 | 0.79 |
| PDS-Base | Y | N | N | N | 0.82 | 0.62 |
| PDS-Abs | Y | Y | N | N | 0.99 | 0.98 |
| PDS-Rob | Y | Y | Y | N | **1.00** | **1.00** |

Table 1: The planning success rate of different models on BabyAI.

Figure 7: (a) and (b) Data efficiency comparison for model learning. We compare three structures with different levels of abstractness. (c) Planning efficiency, measured as the number of expanded nodes for different heuristic computation methods.

Den Oord et al., 2017] to discretize the feature vectors. For all predicates whose output is a latent embedding, we add a vector quantization layer after the encoding layers. We initialize the quantized embeddings by running a K-means clustering over the item feature embedding from a small dataset, and finetune the weights on the entire training dataset for one epoch. We describe this in more detail in the supplementary material. Although our discretizations are inherently lossy on non-Boolean predicate values, since they are only used for search guidance, and not in the forward simulation of the actual actions during planning, this does not affect the correctness of the overall algorithm.

## 3 Experiments

We evaluate PDSketch in two domains: BabyAI, a 2D grid-world and Painting Factory, a simulated tabletop manipulation task. We compare our model with two model-free methods: Behavior Cloning [BC; Bain and Sammut, 1995] and Decision Transformer [DT; Chen et al., 2021]. We implement two DT variants: DT-S with only successful demonstrations, and DT-S+F with successful and failed demonstrations. We use graph neural networks [Gori et al., 2005, Battaglia et al., 2018] as their state encoder.

### 3.1 BabyAI

BabyAI [Chevalier-Boisvert et al., 2019] is an image-based 2D grid-world environment where an agent can navigate around obstacles, pick and place objects, and toggle doors. In this paper, we focus on a specific level of BabyAI, namely *ActionObjDoor*. At this level, the agent navigates within a 7x7 grid. The goals include *go to an X*, *pick up an X*, *open an X*, where X is a noun phrase, such as "blue key". We train all models on environments with 4 doors and 4 objects. The offline dataset $\mathcal{D}$ contains both successful and failure demonstrations obtained by $A^*$ search. We extend PDSketch to interactive data gathering in the supplementary material. Additionally, we test generalization to environments with 6 doors and 8 objects. Since objects may block agents, the agent needs to successfully uncover the underlying dynamics and plan to navigate around them.

We study three PDSketch models with various levels of *locality* and *sparsity* structures built in. The PDS-Base model has no built-in structures: each object is represented as a holistic vector. This falls back to an object-centric transition model [Zhu et al., 2018]. PDS-Abs model that disentangles poses from object appearances. Importantly, it defines a predicate *facing* and uses this concept to write action definitions (e.g., whether a robot move will be blocked by the object it is facing). However, the

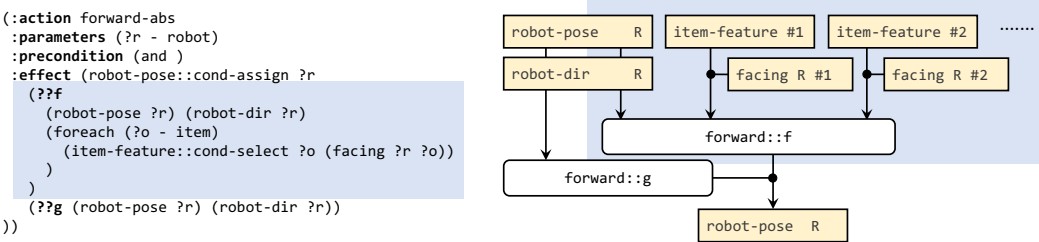

```
(:action forward-abs
 :parameters (?r - robot)
 :precondition (and )
 :effect (robot-pose::cond-assign ?r
   (??f
     (robot-pose ?r) (robot-dir ?r)
     (foreach (?o - item)
       (item-feature::cond-select ?o (facing ?r ?o))
     )
   )
   (??g (robot-pose ?r) (robot-dir ?r))
))
```

Figure 8: The PDSketch definition of the "forward" action and the corresponding computation graph for the PDS-Abs model. The mentioned predicate `facing` is an derived predicate represented as a neural network that is jointly learned (omitted in the graph for brevity).

grounding of *facing* is to be learned. Figure 8 shows the detail definition of the forward action in the PDS-Abs model. PDS-Rob contains predefined rules for robot movements but the object recognition modules are learned. We provide their definitions in the supplementary material.

**Results.** Table 1 shows the results. Overall, PDSketch with more structure (PDS-Abs and PDS-Rob) outperforms baselines by a significant margin. From the performance of PDS-Abs, we see that even a tiny amount of additional structure (e.g., an *ungrounded* predicate "facing") significantly improves the performance, especially when it comes to generalization to more complex environments. See below for a zoomed-in analysis for model learning. Furthermore, we hypothesize that the inferior performance of decision transformer in the mixed training data (Succ+Fail) setting is due to the reward sparsity: the agent only gets reward 1 when it reaches the goal. Thus, the failed demonstrations are generally hard to model as they are irrelevant to the goal specification. In addition, we compare our models with DreamerV2, a state-of-the-art model-based reinforce learning algorithm for image-based environments. Compared with BC and DT, we see DreamerV2 achieves slightly improved performance for the basic task, but does not show stronger generalization to environments with more objects. We hypothesize this is because Dreamer still learns a fixed policy for execution.

**Data efficiency.** Fig. 7a-b shows model learning curves for the three PDSketch models, in the cases where there is a single object in the environment and when there are 4 objects. . When the number of object is small, the environmental dynamics is easier to learn: both Base and Abs have a similar performance. However, when the number of object increases, PDSketch can leverage the inductive bias to learn the dynamics faster. Shown in the figure, even at the end of training, the model PDS-Base has not successfully learned the correct movement dynamics, leading to its inferior performance during generalization to more objects.

**Planning runtime efficiency.** Fig. 7c quantifies the number of nodes expanded by $A^*$ when using different heuristic computations. Specifically, we compare our model with the *blind* heuristic (i.e., the heuristic value of a state is 0 when it satisfies the goal and otherwise 1.) and a GNN-parameterized heuristic learned from successful demonstrations. All results are based on the PDS-Abs model. First, learning-based heuristics (GNN) shows improvement over the baseline "blind" heuristic, especially when the task is easy (requiring a small number of expansions). Second, the heuristic derived from PDSketch significantly improves the search efficiency. At the success rate of 0.8, our method is 5 times more efficient than learning-based heuristics.

## 3.2 Robot Painting

Finally, we extend the framework to a tabletop manipulation task, built based on the tabletop environment of Zeng et al. [2020]. There are three zones, several bowls, and several blocks on the table. The robot can use its suction gripper to pick-and-place objects into a designated location. Blocks have 8 possible colors. Placing objects in a bowl will paint the object to be the same color as the bowl. The task is to paint the blocks and organize them in the target brown box. Our training-time goal requires the robot to paint-and-place two objects. The goal contains their colors and their relationship (e.g., a *pink* block is *left of* a *yellow* block. Demonstration are collected using hand-crafted oracle policies following Zeng et al. [2020]. The offline dataset contains only successful trajectories. There are two built-in actions that the robot can execute. Fig. 9 shows the computation graph derived from PDSketch.

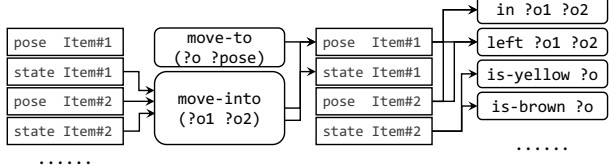

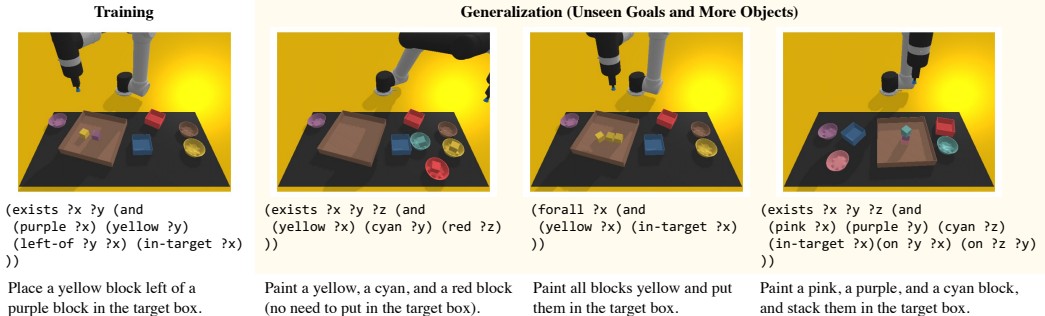

| Model | Succ. Rate @100 | Succ Rate @1000 |
|---|---|---|
| BC | 0.70 | 0.95 |
| DT | 0.56 | 0.97 |
| PDS | **0.91** | **0.99** |

Figure 9: Our encoding of the painting factory domain. Each object has a pose and a state. Action `move-to` directly changes the object pose. `move-into` moves an object into a container, which possible changes the state (e.g., color) of the object. Goals are represented using relational predicates.

Table 2: Model performance on Painting Factory, measured as plan success rate. The number after @ is the number of demonstration trajectories. PDSketch models show stronger data efficiency than baselines.

**Training**

```
(exists ?x ?y (and
 (purple ?x) (yellow ?y)
 (left-of ?y ?x) (in-target ?x))
))
```
Place a yellow block left of a purple block in the target box.

**Generalization (Unseen Goals and More Objects)**

```
(exists ?x ?y ?z (and
 (yellow ?x) (cyan ?y) (red ?z))
))
```
Paint a yellow, a cyan, and a red block (no need to put in the target box).

```
(forall ?x (and
 (yellow ?x) (in-target ?x))
))
```
Paint all blocks yellow and put them in the target box.

```
(exists ?x ?y ?z (and
 (pink ?x) (purple ?y) (cyan ?z)
 (in-target ?x)(on ?y ?x) (on ?z ?y)
))
```
Paint a pink, a purple, and a cyan block, and stack them in the target box.

Figure 10: From only one training task (left), a model specified in PDSketch learns primitive concepts including object colors and spatial relations. They can be used in planning for unseen tasks given new first-order logic descriptions of the goal. The natural language descriptions are shown for readability.

**State and action space.** The state space in the painting factory is composed of a list of objects, and a list of containers. Both objects and containers are represented as a tuple of a 3D xyz location, and a image crop. The image crop is generated by first computing the 2D bounding box of the object in the camera plain, cropping out the image patch, and resizing it to 32 by 32. The action space contains two primitives. First, `move-into` is an action defined for each pair of item and container. It changes the pose of the item (now the item will be inside the container). When the item is placed into a bowl, it will be painted into the same color as the bowl. The second action takes an object and a 3D location as input, and moves the object directly to a designated position (ignoring the rotation).

**Results.** Table 2 shows the planning success rate *on the training task* with different amounts of training data. Overall, PDSketch is more data efficient than both baselines and achieves strong overall performance in this task. Much more importantly, Fig. 10 shows our *generalization to novel task specifications*, which involve more objects or new specifiers (e.g., `forall`). Our model generalizes directly to these novel scenarios *without any additional training*. The quantitative performance for these three generalization tasks are: 0.99, 0.98, and 0.87, respectively, measured as success rate after executing the plan. The last task has lower success rate because stacking objects may fail due to controller and physical noises. Future work may consider building closed-loop controller that can recover such failures. We specify implementation details in the supplementary material.

## 4   Related Work

Integrated model learning and planning is a promising strategy for building robots that can generalize to novel situations and novel goals. Specifically, Chiappa et al. [2017], Zhang et al. [2021], Schrittwieser et al. [2020] learn dynamics from raw pixels; Jetchev et al. [2013], Pasula et al. [2007], Konidaris et al. [2018], Chitnis et al. [2021], Bonet and Geffner [2020], Asai and Muise [2020], Silver et al. [2021] assume access to the underlying factored states of objects, such as object colors and other physical properties. Our work bridges the gap between two groups: we do not assume pre-factored state representations given as the input, but learn to ground different factors of object states. More importantly, instead of relying on off-the-shelf models for predicting pixels or for

learning first-order rules in symbolic domains, our work considers how human-programmed locality and sparsity structures can improve the model learning and planning efficiency.

Our model learns an object-factored transition model, which generally falls into the category of learning object-oriented MDPs [OO-MDPs; Guestrin et al., 2003, Diuk et al., 2008]. OO-MDPs have been applied to neural network-based representations for visual domains [Walsh, 2010, Kansky et al., 2017, Zhu et al., 2018, Xia et al., 2019, Veerapaneni et al., 2020] and textual game domains [Liu et al., 2021]. Our model leverages object-centric representations to encode permutation-invariant structures. Furthmore, we exploit fine-grained local and sparse structures in model learning.

The objective of model learning and planning is to obtain a goal-conditioned policy [Kaelbling, 1993b,a]. While many others have studied model-free [Dayan and Hinton, 1992, Schaul et al., 2015] or hybrid model-free and model-based approaches [Pong et al., 2018, Nasiriany et al., 2019], in this paper, we focus on learning *structured* models from data and leveraging domain-independent planners for planning in latent representations. Such structured models support novel goal specifications via a first-order logic language, and domain-independent heuristics to accelerate search. Another alternative approach towards learning and planning is to learning to predict subgoals that needs to be achieved before other goals [Xu et al., 2019]. However, their work assumes that all preconditions and effects can be represented in a predefined symbolic language, and there are controllers for achieving individual subgoals. By contrast, PDSketch supports generic neural-network-based representations for predicates and action effects.

Our framework PDSketch combines model definition in a structured language (e.g., first-order logic) and neural network learning. It is closely related to the idea of neuro-symbolic programming and differentiable programming for relational reasoning [Manhaeve et al., 2018, Riegel et al., 2020, Huang et al., 2021] and policy learning [Sun et al., 2019]. Our language PDSketch borrows ideas from earlier work on "soft" execution of logical formulas but works on planning domains.

## 5 Conclusions and Limitations

PDSketch supports flexible and effective specification of *locality* and *sparsity* structures of environment transition models. Leveraging these structures enables more data-efficient learning, compositional generalization to novel goal specifications and environmental states, and also domain-independent heuristics that accelerates performance-time planning. Limitations we hope to address include the lack of hierarchy and the inability of the method to discover novel factorizations.

In terms of societal impact, PDSketch suggests a hybrid method for building intelligent robots, by combining human programs to specify high-level structures with learning, to reduce the amount of data and computation required for learning complex and long-horizon behaviors. It also enables more modularized systems: users can specify new tasks using the predicates and actions that have already been defined more easily than in unstructured approaches. However, PDSketch may require more expertise (or, at least, a different type of expertise) from programmers than other approaches.

**Acknowledgement.** We thank all group members of the MIT Learning & Intelligent Systems Group for helpful comments on an early version of the project. This work is in part supported by ONR MURI N00014-16-1-2007, the Center for Brain, Minds, and Machines (CBMM, funded by NSF STC award CCF-1231216), NSF grant 2214177, AFOSR grant FA9550-22-1-0249, ONR grant N00014-18-1-2847, the MIT Quest for Intelligence, MIT–IBM Watson Lab. Any opinions, findings, and conclusions or recommendations expressed in this material are those of the authors and do not necessarily reflect the views of our sponsors.

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
