# Supplementary Material for
# PDSketch: Integrated Planning Domain Programming and Learning

**Jiayuan Mao**[1]    **Tomás Lozano-Pérez**[1]    **Joshua B. Tenenbaum**[1,2,3]    **Leslie Pack Kaelbling**[1]

[1] MIT Computer Science & Artificial Intelligence Laboratory
[2] MIT Department of Brain and Cognitive Sciences
[3] Center for Brains, Minds and Machines

We have released an alpha version of our code at http://pdsketch.csail.mit.edu/. The rest of the supplementary material is organized as the following. First, in Section A, we provide additional results on data efficiency and on-policy learning for the BabyAI environment studied in the paper. Then, in Section B, we present an example-based formal introduction of the PDSketch language. Next, in Section C, we describe the implementation of our domain-independent heuristic. We also discuss why sparsity and locality structures are particularly important in inducing domain-independent heuristics. Finally, in Section D, we present our experiment setups. Both Section C and Section D are presented based on the language definition in Section B.

## A    Additional BabyAI Results

In this section, we supplement two additional results on the BabyAI environment. First, in Section A.1, we present the data efficiency study of different models in terms of their performance-time success rate. We also make variance analysis of different models across different runs. Next, in Section A.2, we present how our model can be integrated with on-policy learning methods to make active data gathering.

### A.1    Data Efficiency and Variance Analysis

We further quantify the data efficiency of different models in terms of their performance-time success rate. Note that this is different from the Figure 7 in the main paper, where we have studied the data efficiency in terms of model learning accuracy. In this section, we show that, 1) compared to model-free baselines, integrate model learning and planning does improve the overall data efficiency in the domain we consider, and 2) the same is true for different PDSketch models with different levels of abstractions and prior knowledge.

Concretely, we generate three datasets, with 100 demonstration episodes, 1000 episodes, and 10,000 episodes, respectively. We train the model on different datasets until converge, and evaluate their performance. We have repeated all experiments three times. The reported value are the average score and the standard deviation.

Results are summarized in Table 1. Here we summarize our findings as the following.

1. First, in general, behavior cloning and decision transformer achieve similar performance on this task. When the amount of training data is small, the model perform noticeably worse than model-based methods.

2. The base model, even if trained with a large amount of data, struggle to capture the robotic movement transitions when facing obstacles. This results in that the performance gets stuck at around 80%.

36th Conference on Neural Information Processing Systems (NeurIPS 2022).

| Model | 100 episodes | 1000 episodes | 10000 episodes |
|---|---|---|---|
| BC | $0.24 \pm 0.01$ | $0.24 \pm 0.01$ | $0.39 \pm 0.03$ |
| DT | $0.23 \pm 0.02$ | $0.24 \pm 0.01$ | $0.30 \pm 0.05$ |
| PDS-Base | $0.04 \pm 0.00$ | $0.11 \pm 0.01$ | $0.80 \pm 0.07$ |
| PDS-Abs | $0.04 \pm 0.02$ | $0.98 \pm 0.03$ | $\mathbf{0.99} \pm 0.01$ |
| PDS-Rob | $\mathbf{0.72} \pm 0.09$ | $\mathbf{0.99} \pm 0.01$ | $\mathbf{0.99} \pm 0.01$ |

Table 1: The planning success rate for different models using different amount of training data.

3. Compared to recognizing object properties, a significant amount of data is required to learn the transition model. When the transition model is given (in PDS-Rob), learning the visual recognition models in this simple grid-world domain is very data-efficient: using only 100 episodes, the model successfully solves 70% of the tasks.

## A.2 On-Policy Learning

Our system can also be integrated into an on-policy learning setting. This is compatible with the standard model-based reinforcement learning setting. Specifically, recall that in an interactive learning setting, the system receives the current state and a goal expression. The agent itself should decide the next action to take. In the PDSketch case, we run the planner based on the current model parameters $\theta$ to generate a plan. We follow the plan in the simulator and collect the resulting trajectory as well as the "done" signals from the environment. Finally, we update the model parameter $\theta$ using the newly collected data. Although our model may leverage offline data, to keep the algorithm simple, we do not implement data reusing. Within 10,000 episodes, our model PDS-Rob is able to reach 0.99 performance-time success rate. Comparatively, the reported performance of Proximal Policy Optimization (PPO) [Schulman et al., 2017, Chevalier-Boisvert et al., 2019] needs 200,000 episodes.

We found that direct application of the on-policy algorithm on models with less inductive biases (i.e., PDS-Abs and PDS-Base) does not yield successful results. The success rate stuck around 6%. This is primarily because of their failure in exploration: direct application of the planner generates successful trajectories at a very low probability. This can be potentially alleviated by adding random exploration factors or replanning when the agent fails to reach the goal.

Note that, since our model solely focuses on representing and learning the model, it is possible to integrate our framework with other model-based reinforcement learning algorithms, such as joint model and policy learning [Schrittwieser et al., 2020], or world-model-based reinforcement learning [Hafner et al., 2021]. We leave these extensions as a future work.

## B  PDSketch Language

In this section, we detail the design, the syntax, the semantics, and the implementation of the PDSketch definition language. We present it in two parts: the predicate and action definition (Appendix B.1), and PDSketch expressions (Appendix B.2).

### B.1  Predicate and Action Definition

PDSketch is based on the Planning-Domain Definition Language [Fikes and Nilsson, 1971, Fox and Long, 2003], which is a language specialized and derived from LISP. We choose a LISP-style language for two important reasons. First, compared to other procedure- or object-centric languages such as Python, LISP features an easy definition of first-order logic, especially for recursively defined rules. Second, a significant amount of planning domains have been written in PDDL. Extending the language reduces the learning cost for programmers and also make old PDDL definitions easily reusable in PDSketch.

For readers who are familiar with PDDL, a PDSketch file is a domain definition file (in contrast to a problem definition file). PDSketch only contains the definition of predicates and actions in the domain. It does not contain problem-specific information, such as the current state of the world, or the goal specification.

Thus, a PDSketch file contains three parts. A PDSketch definition, at the highest level, takes the following form:

```
definition:      "(" "define" "domain" domain-name-def content-def* ")"
domain-name-def: "(" "domain" string ")"
content-def:     type-def | predicate-def | derived-def | action-def
```

An empty domain file can be written as:

```
(define domain
  (domain my-domain-name)
)
```

This defines a domain with name "my-domain-name," with no predicates and actions defined.

A `content-def` can be either a type definition, a predicate definition, or an action schema definition. First, type definition:

```
type-def:        "(" ":types" single-type-def ")"
single-type-def: type-name-list - base-type
type-name-list:  type-name+
type-name:       string
base-type:       type-name | prim-type | vector-type
prim-type:       "bool" | "int64" | "float32"
vector-type:     "vector" "[" prim-type (, int)? "]"
```

Below, we will be using the BabyAI world as an example, the example type definition of the BabyAI domain is:

```
(:types
```

```
  robot item - object
  pose       - vector[float32, 2]
  direction  - vector[int64, 1]
)
```

Specifically, a type definition contains multiple lines, each line being a pair of a type name list, and a base type. There are two types of types in PDSketch: object type and value type. Intuitively, object type refers to a concrete object in the world, while a value type usually denotes the return type of a feature function over the objects. Currently, we do not support the full hierarchical structure of type definitions: each type either inherits "object", which indicates this is an object type, or inherits a primitive value type (Boolean, integer, or floating-point numbers), or a vector type. In a vector-type definition, the second argument (e.g., the 2 in vector[float32, 2]) denotes the dimension of the vector. This can be omitted. Overall, the definition above defines four types: an object type called "robot," an object type called "item," a value type that is a 2D vector, named "pose," and a value type that is a 1D vector of integer (i.e., a single integer), named "direction."

The predicate-def and derived-def are jointly used to define predicates. A predicate can be either a predicate directly observable from the environment, or a "derived" predicate, which is a predicate whose value is computed based on the value of other predicates. We first show the grammar definition.

```
predicate-def:        "(" ":predicates" single-predicate-def ")"
single-predicate-def: "(" predicate-name kwargs? variable-list ")"
predicate-name:       string
kwargs:               "[" kwarg "]"
kwarg:                kwarg-key "=" kwarg-value
kwarg-value:          "\"" string "\"" | int | bool | float | base-type
variable-list:        typed-variable*
typed-variable:       variable-name "-" type-name
variable-name:        string
```

An example definition of some input predicates is the following:

```
(:predicates
  (robot-pose      [return_type=pose]            ?r - robot)
  (robot-direction [return_type=direction]       ?r - robot)
  (item-pose       [return_type=pose]            ?o - item)
  (item-image      [return_type=vector[float32]] ?o - item)
)
```

In this case, we have defined four predicates: a 2D position of the robot, the direction that the robot is facing, the 2D position of the item, and finally, an image representing a local crop of the item.

Based on the input predicates, users can define many "derived" predicates, whose values are automatically computed based on other input and derived predicates. The formal syntax is:

```
derived-def: "(" ":derived" single-predicate-def expr ")"
```

in which `expr` is a syntax for expressions, which we will detail later. As an example, we look at the definition of several item property-related predicates.

```
(:derived (item-feature [return_type=vector[float32, 64]] ?o - item)
  (??f (item-image ?o))
)
```

This definition defines a predicate named `item-feature`. It takes an item as its argument, and returns a 64-dimensional vector embedding associated with the object. Its expression is (`??f (item-image ?o)`). That is, an unknown mapping from the input image of the item to a vector embedding. This definition will implicitly define a function whose name is `derived::item-feature::f`. This function has no default implementation. The programmer, after loading the PDSketch definition, should register the corresponding implementation of this function. For example, one option is to associate this function with a convolutional neural network (CNN) that takes the image crop of each object `?o` and computes the corresponding item feature. Note that, in this definition, we are also implicitly defining a *parameter sharing* strategy: for all objects in the environment (independent of its absolute index), we will be applying the same CNN identically to them. We will detail possible expressions PDSketch supports later.

Based on this `item-feature` definition, users can define several classifiers that will be useful in defining goal predicates.

```
(:derived (is-red  ?o - item) (??f (item-feature ?o)))
(:derived (is-blue ?o - item) (??f (item-feature ?o)))
; ... more colors
(:derived (is-ball ?o - item) (??f (item-feature ?o)))
(:derived (is-door ?o - item) (??f (item-feature ?o)))
; ... more shapes
(:derived (is-open ?o - item) (??f (item-feature ?o)))  ; for doors
```

By default, when the return type is not specified, the function returns Boolean values, which is consistent with the original PDDL syntax.

In BabyAI, items that the agent is holding has a special 2D position which is [-1, -1]. Thus, whether the agent is holding an object can be classified by

```
(:derived (robot-holding ?r - robot ?o - item)
  (??f (item-pose ?o))
)
```

Finally, we want to define a function indicating whether the robot is facing the object,

```
(:derived (robot-facing ?r - robot ?o - item)
  (??f (robot-pose ?r) (item-pose ?o))
)
```

In its expression, we are defining an unspecified function `??f`, which takes two arguments, the robot position, and the item position. With all these predicates, we can now define the tasks. Recall that the ActionObjDoor environment has the following three tasks:

**go to a red box:**

```
(exists (?o - item) (and
  (robot-facing agent ?o) (is-red ?o) (is-box ?o)
))
```

**pick up a red box:**

```
(exists (?o - item) (and
  (robot-holding agent ?o) (is-red ?o) (is-box ?o)
))
```

**open a red door:**

```
(exists (?o - item) (and
  (is-red ?o) (is-door ?o) (is-open ?o)
))
```

In the first two goal specifications, we are using a name constant "agent" to denote the robot that we are considering.

Next, we are going to define actions. There are four actions the agent can perform. The syntax for action definition is:

```
action-def: "(" ":action" (action-name kwargs)
  ":parameter" "(" variable-list ")"
  ":precondition" expr
  ":effect" expr
")"
```

An action definition is composed of a name, a list of parameters, a precondition, and an effect. The parameter are the arguments to the action. The precondition is a Boolean output value, indicating the

situation where this action can be executed. Note that, the precondition is not part of the transition model of the environment. Instead, it's a property that is associated with the underlying policy of this action. We will come back to this point when we discuss concrete examples. The effect is an expression that change the state variables according to certain rules.

Concretely, we start with a simple action: "lturn," which models the behavior that the robot turns left.

```
(:action lturn
 :parameters (?r - robot)
 :precondition (and )
 :effect (assign (robot-direction ?r) (??f (robot-direction ?r)))
 ; could also be written as:
 ; :effect (robot-direction::assign ?r (??f (robot-direction ?r)))
)
```

This definition has defined an action named "lturn," it takes a single parameter "?r" which indicates the robot that we want to control. It has an "empty" precondition, indicating that this primitive action can be executed anytime. Meanwhile, the effect is to change the "robot-direction" state variable associated with the robot `?r`. The rule is a based on an (unspecified) function that takes the current robot facing direction and returns the new direction: `(??f (robot-direction ?r))`.

With first-order logic formula, one can define more complex actions. One good example is the definition of the "forward" action. In BabyAI, the forward action takes the following rule. When there is no object facing the agent, the agent can move forward. When there is an object facing the robot, the robot may be blocked by the object, in which case the robot position will not change. Note that not all map items will block robots' movement. Which type of objects will block the robot is to be learned by the learning algorithm. In this case, we can define the robot action in an intuitive way as the following:

```
(:action forward
 :parameters (?r - robot)
 :precondition (and )
 :effect (robot-pose::assign ?r (??f
   (robot-pose ?r)
   (robot-direction ?r)
   (foreach (?o - item)           ; enumerate all items in the map
     (when (robot-facing ?r ?o)   ; if the robot is facing this item
       (item-feature ?o)          ; consider the item-feature of ?o
     )
   )
 ))
)
```

We now break down this definition into pieces. First, this is an action that takes only one argument (the robot "?r"). Similar to the case of "lturn," this action does not have a precondition, indicating that the action can be applied in any situations. The effect definition for this action is slightly more complex. First, the action changes the state variable (`robot-pose ?r`). The new value is computed based on three things, the current `robot-pose`, the current `robot-direction`, and the feature of all objects `?o` that the robot is facing. Here, we are using two special keywords, `foreach` and `when`. The first one iterates over all items in the environment, and the second one selects the feature of object `?o` if the condition is satisfied. Importantly, in this case, the corresponding function `f` should be implemented as a variable-length-input function. That is, depending on how many objects are selected, the number of input features will change. In our implementation, this function is implemented as a graph neural network (GNN). Of course, if the programmer knows more about the underlying transition model, they can specify more detailed structure, such as the following:

```
; define a helper derived predicate.
(:derived (is-obstacle ?o - item) (??f (item-feature ?o)))
(:action forward-detail
 :parameters (?r - robot)
 :precondition (and )
 :effect (when
   ; when there is no item ?o such that
   ;   ?o is an obstacle and the robot is facing ?o
   (not (exists (?o - item) (and (is-obstacle ?o) (robot-facing ?r ?o)) ))
   ; the robot pose will move forward and the new pose
   ;   will be computed by the current pose and the facing direction.
   (assign (robot-pose ?r)
     (??f (robot-pose ?r) (robot-direction ?r))
   )
 )
)
```

Similarly, we can have the following definition for "the robot executes picking up."

```
; define a helper predicate indicating whether the object
;   can be picked up.
(:derived (can-pickup ?o) (??f (item-feature ?o)))
(:action pickup
 :parameters (?r - robot)
 :precondition (and )
 :effect (foreach (?o - item)
   (when (and (robot-facing ?r ?o) (can-pickup ?o) )
     (assign (item-pose ?o)
       ; a dummy function, that should be implemented to return
       ;   [-1, -1], indicating the item is in robot's inventory.
       (??f )
     )
   )
 )
)
```

Overall, the definition language allows users to encode various kinds of prior knowledge about the transition model into the representation. Meanwhile, it does not require human programmers to fully specify the details, especially the recognition and classification problems. The learning algorithm will follow the structure and learns the missing pieces.

Importantly, depend on the knowledge that the programmer has about the environment, such definition can be very detailed, or very abstract. At an extreme case, we can have the following definition:

```
(:action mysterious-action
 :parameter (?r - robot)
 :precondition (and )
 :effect (and
   (robot-pose::assign ?r (??f1
     (robot-pose ?r) (robot-direction ?r) (item-pose ??) (item-feature ??)
   ))
   (robot-direction::assign ?r (??f2
     (robot-pose ?r) (robot-direction ?r) (item-pose ??) (item-feature ??)
   ))
   (foreach (?o - item) (item-pose::assign ?o (??f3
     (robot-pose ?r) (robot-direction ?r) (item-pose ??) (item-feature ??)
   )))
   (foreach (?o - item) (item-feature::assign ?o (??f4
     (robot-pose ?r) (robot-direction ?r) (item-pose ??) (item-feature ??)
   )))
 )
)
```

Here we are using a syntax sugar: the notation `(item-pose ??)` is equivalent to `(foreach (?x - item) (item-pose ?x))`. That is, this function takes the pose of all items into consideration. Note that, in this case, we have *no* structure built in to the system: the action can change any state variable, and the change depends on all other variables in the state representation. In this case, depending on the actual implementations of `f1` to `f4`, this transition model can be implemented in any form, such as a graph neural network (thus an object-centric transition model).

**Remark: Preconditions *vs*. conditional effects.**  There are two seemingly similar ways to define the condition under which an action takes effect: preconditions and conditional effects. To better understand the different between these two options, consider the following two examples:

```
(:predicates
  (p ?o - item)
  (q ?o - item)
)
(:action op1
 :parameters (?o - item)
 :precondition (p ?o)
 :effect (q:assign (??f (q ?o)))
)
(:action op2
 :parameters (?o - item)
 :precondition (and )
 :effect (when
   (p ?o)
   (q:assign (??f (q ?o)))
 )
)
```

The key difference is that, a "precondition" is the property associated with the corresponding low-level controller for the action, that is $\pi_{op1}$. Its semantics is that, the $\pi_{op1}$ can only be executed when (p ?o) returns true. It does not specify what will happen when we execute $\pi_{op1}$ in such a situation. In contrast, the semantics of the second definition is that, the corresponding $\pi_{op2}$ can be executed under any circumstances. However, only if the condition for the conditional effect is true (i.e., (p ?o), the state value for q will be changed.

Thus, preconditions and conditional effects should be learned from different signals. Preconditions should be learned from the signal from the execution of the corresponding controller. More specifically, the controller $\pi_{op1}$ should return a Boolean value indicating whether the controller can be executed. By contrast, the conditional effects should be learned from the outcome of the execution of the controller $\pi_{op2}$.

Note that, the conditional effect formulation is the one that fits most of the concurrent environment setups for reinforcement learning. That is, the defined actions correspond to the primitive actions that the robot can execute. Thus, usually all actions are always applicable at any state (i.e., the precondition of all actions are empty).

## B.2 Expressions

PDSketch has the following built-in operations.

**Propositional logic operations: `and`, `or`, `not`, and `implies`.** The propositional logic operations take the following syntax:

```
and-expr-def:     "(" "and"      expr* ")"
or-expr-def:      "(" "or"       expr* ")"
not-expr-def:     "(" "not"      expr  ")"
implies-expr-def: "(" "implies" expr expr ")"
```

The semantics of these operators follows the convention of Boolean operations: conjunction, disjunction, negation, and implication. Their implementation are based on the differentiable Gödel t-norms: $\text{not}(p) = 1 - p$, $\text{and}(p_1, p_2) = \min(p_1, p_2)$, $\text{or}(p_1, p_2) = \max(p_1, p_2)$, and $\text{implies}(p_1, p_2) = \max(1 - p_1, p_2)$.

There are two conventional notations for Boolean operators used in the definition of action effects. First, when an action has multiple effects, they will be written within a big `and` block. For example, the following definition:

```
(:action set-pqr
 :parameters (?r - robot)
 :precondition (and )
 :effect (and
   (assign (p ?r) (??f))
   (assign (q ?r) (??f))
   (assign (r ?r) (??f))
 )
)
```

Meanwhile, when a Boolean predicate is directly written in the effect, its semantics is that this Boolean state variable will be set to true. Similarly, when we write `(not (t ?r))` where `t` is a Boolean predicate, it is equivalent to `(assign (t ?r) true)`.

**Boolean quantification: `forall` and `exists`.** Similarly, the quantification operations are define as the following:

```
forall-expr-def: "(" "forall" (typed-variable) expr ")"
exists-expr-def: "(" "exists" (typed-variable) expr ")"
```

As an example, the following expression:

```
(exists (?o - item) (is-red ?o))
```

computes whether there is an item in the environment that is red.

Similarly, the implementation of Boolean quantifiers are: $\forall x.p(x) = \min_x p(x)$ and $\exists x.p(x) = \max_x p(x)$.

**Assignment.** There is only one assignment operation

```
assign-expr-def: "(" "assign" "(" predicate-name variable-list ")" expr ")"
```

The first term is the target state variable, while the second term is the expression of the value. For example, (assign (p ?r) (??f)) means that the value of (p ?r) will be assigned to the return value of function f.

**Foreach and condition.** There are two special operators: for-each and condition.

```
foreach-expr-def: "(" "foreach" (typed-variable) expr ")"
when-expr-def:    "(" "when" expr expr ")"
```

When they are used in a precondition or an expression (in contrast to being under the effect definition), they means:

- `foreach`: it selects all objects of the specified type. See the previous definition of action `forward` as an example.
- `when`: it specifies the scenario in which a state variable is relevant to the computation. See the previous definition of action `forward` as an example.

When they are used in effect definitions, they means:

- `foreach`: it applies the effect formula to objects of the specified type. See the previous definition of action `mysterious-action` as an example.
- `when`: it indicates a conditional effect. The effect will be applied if and only if the condition is true. See the previous definition of action `mysterious-action` as an example.

**Syntax sugars.** PDSketch includes the following syntax sugars:

(p::assign ?a1 ?a2 ...  VALUE) is equivalent to
(assign (p ?a1 ?a2 ...)  VALUE).

(p::cond-assign ?a1 ?a2 ...  COND VALUE) is equivalent to
(when COND (assign (p ?a1 ?a2 ...)  VALUE)).

(p::cond-select ?a1 ?a2 ...  COND) is equivalent to
(when COND (p ?a1 ?a2 ...)).

**Blank notation ??.** The **??** operation can be used to replace any expressions. It takes the following syntax:

```
slot-expr-def: "(" "??" predicate-name kwargs expr* ")"
```

For example, the definition: (??f (item-feature ?o)) defines an function that will be externally implemented. It has name f. It takes only one input, which is (item-feature ?o).

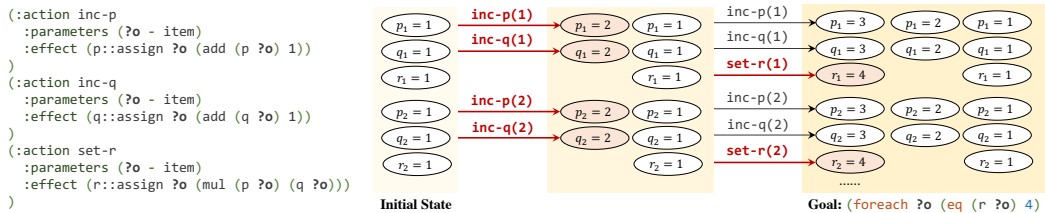

```
(:action inc-p
  :parameters (?o - item)
  :effect (p::assign ?o (add (p ?o) 1))
)
(:action inc-q
  :parameters (?o - item)
  :effect (q::assign ?o (add (q ?o) 1))
)
(:action set-r
  :parameters (?o - item)
  :effect (r::assign ?o (mul (p ?o) (q ?o)))
)
```

Figure 1: A graphical illustration of the hFF heuristics. The goal is to set the r value of all objects to 4. The heuristic value hFF = 6, which is the number of selected operators (highlighted in red).

## C   Domain-Independent Heuristic

In this paper, we focus on the hFF heuristic. Based on relaxed operators, it heuristically selects a set of operators that should be applied to accomplish the goal. Shown in Fig. 1, hFF sequentially tries to apply operators that yield novel values of the features. Once the goal condition is satisfied, it back-traces the used operators. Note that such computation is general and applies to domains with arbitrarily complex feature dependencies and number of objects. We present details of our implementation in the supplementary. However, such heuristics does not naturally work with vector embeddings and neural network predictors. Next, we talk about how to "compile" trained neural features and modules into hFF-compatible representations.

In this paper, we explore two strategies for performing this reduction. In the first, we simply change all the action models so that the result is to set all of the predicate instances that they effect to have a special value called *optimistic*, which is assumed to satisfy any goal test on that predicate instance. (So, for example, after we have placed an object somewhere, we immediately "believe" it could be anywhere.) An alternative strategy, which is more computationally complex but yields "tighter" (and therefore more helpful) heuristic values, is to explicitly discretize the value space of the predicates and directly reduce to the standard discrete case. Both of these strategies are described in more detail in the supplementary material. It is important to note that these methods are inherently lossy on non-Boolean predicate values, but since they are only used for search guidance, and not in the forward simulation of the actual actions during planning, they do not affect the correctness of the overall algorithm.

**Optimistic compilation: leveraging locality.**   The first approach, optimistic compilation (OPT), compiles each non-Boolean state variable, e.g., `(color ?o)`, into a boolean predicate `color-opt`. Any operator that changes the value of `color` (e.g., `press-button`) will set `color-opt` to true. Meanwhile, any Boolean expressions, such as `(?? (color item#1))` will return true as long as `(color-opt item#1)` is true. Intuitively, any operator that changes feature values will make the change *optimistically*.

The optimistic compilation ignores the fine-grained structure inside each state variables, but does leverage the object-level locality. For example, in the case of `(forall ?o (eq (r ?o) 4)`, it will select two operators `set-r(1)` and `set-r(2)`, leading to hFF = 2.

**And-Or compilation.**   The And-Or compilation (AO) works in two steps. First, it discretizes all continuous state variables (poses, etc.) into a designated number of bins (e.g., 128). Next, for each externally-defined functions, it learns a first-order decision tree to approximate the computation (e.g., to approximate a neural network). Concretely, we use VQVAE [Van Den Oord et al., 2017] to discretize the feature vectors. For all predicates whose output is a latent embedding, e.g., the `(item-feature ?o)` we introduced in our earlier example, we add a vector quantization layer after the encoding layers. We initialize the quantized embeddings by running a K-means clustering over the item feature embedding from a small dataset, and finetune the weights on the entire training dataset for one epoch.

For all predicates whose inputs and outputs are both continuous parameters. We use the FOIL algorithm, the first-order decision tree learning algorithm [Quinlan, 1990] to extract first-order decision rules. We need to use FOIL instead of a plain (propositional) decision tree learning algorithm

because the input to decision functions may have a variable number of objects. Here we are showing some examples of the quantized predicates:

```
(is-red ?o - item) = (or
  (item-feature@3  ?o)
  (item-feature@14 ?o)
  (item-feature@16 ?o)
  (item-feature@10 ?o)
)
```

Below we are showing a slightly more complex example, corresponding to the learned rule for action "forward."

```
(((robot-pose ?r - robot) <- (SAS
  37 <- (or
  (and
    (not (robot-direction@2 ?r - robot))
    (not (robot-direction@3 ?r - robot))
    (not (robot-direction@1 ?r - robot))
    (robot-pose@37 ?r - robot)
  )
  (and (robot-direction@3 ?r - robot) (robot-pose@45 ?r - robot))
  (and (robot-pose@36 ?r - robot) (robot-direction@0 ?r - robot))
  (and (robot-pose@29 ?r - robot) (robot-direction@1 ?r - robot))
)
  4 <- (and (robot-pose@12 ?r - robot) (robot-direction@3 ?r - robot))
  40 <- (and (robot-pose@41 ?r - robot) (robot-direction@2 ?r - robot))
  21 <- (and (robot-pose@29 ?r - robot) (robot-direction@3 ?r - robot))
  32 <- (and (robot-pose@33 ?r - robot) (robot-direction@2 ?r - robot))
  3 <- (and
    (robot-pose@11 ?r - robot) (robot-direction@3 ?r - robot))
    (and
      (not (robot-pose@27 ?r - robot))
      (not (exists (_t0 - item) (and (robot-is-facing ?r - robot _t0 - item)
                                     (item-feature@9 _t0 - item))))
      (not (exists (_t0 - item) (and (robot-is-facing ?r - robot _t0 - item)
                                     (item-feature@0 _t0 - item))))
      (not (exists (_t0 - item) (and (robot-is-facing ?r - robot _t0 - item)
                                     (item-feature@6 _t0 - item))))
      (not (exists (_t0 - item) (and (robot-is-facing ?r - robot _t0 - item)
                                     (item-feature@11 _t0 - item))))
; ... more rules omitted.
```

Note in the last section of the shown rules, the decision rule is written in first-order logic, where we need to quantify over objects.

The AO compilation further leverages the fine-grained structures of state variables. For example, the real value of features p, q, and r will be discretized into bins, such as p=1, p=2, q=1, q=2. A decision tree will be used to represent the transition model. Thus, the resulting model will support fine-grained simulation and back-tracing.

**The role of sparsity and locality structures in heuristics.** Although the discretization and heuristic computation itself does not assume any particular structure of the action definitions, their performance does rely on the programmed sparsity and locality structures, in particular, the "factorization" structure of the feature representation.

Specifically, without factorization, the entire state is described with one single vector embedding. This introduces a exponential number of possible states of the state. For example, let's assume an object state can be factorized into its x and y position (7 possible states per dimension), the color (6 possible states), and the shape (4 possible shapes). With factorization, we only need 24 possible state codes to represent each object. However, without factorization, we need $7 \times 7 \times 6 \times 4 = 1176$ states. Recall that, during the computation of heuristics, we will be solving a relaxed version of the planning program. If there is no factorization, the search problem reduces to a plain path-finding problem in the graph induced by the connectivity between these states, which is very inefficient to solve.

# D  Experimental Setup.

In this section, we detail the environment setup and domain definition of two environments: BabyAI and Painting Factory.

## D.1  BabyAI

**Setup.**  Following the original BabyAI setup, we use a 7 by 7 grid as the physical world. The agent is initially positioned at the center, all other object locations are randomly selected.

**Offline data.**  The data contains both successful demonstrations and unsuccessful demonstrations. For successful ones, we use the grid-world $A^*$ search to generate the optimal trajectory from the original agent position to the target. For unsuccessful demonstrations, we first choose an incorrect object, approach the selected object, and then runs 5 steps of random walk.

**Baselines.**  For all baselines and our models, we use the following ways to encode object states, robot positions, and facing directions.

1. object state: following the FiLM model presented in Chevalier-Boisvert et al. [2019], we use an integer embedding for object states.
2. object position and robot position: we directly use the raw value as the input to the neural networks. We have tried to use embedding based methods to encode the values, but have seen consistently worse performance.
3. robot facing direction: there are four directions. We use a 4 learnable embeddings for them.

Thus, the input to the baselines are: 1) the robot state embedding, 2) the task goal specification, encoded as the concatenation of three word embeddings: the verb, the adjective, and the noun, and 3) the object state embeddings.

For both baselines BC and DT, we use a two-layer graph neural network to encode the world state into a 128-dimensional vector embedding. For BC, we use a single linear layer to predict the action to take. For DT, we use a transformer layer to aggregate the history (especially encoding the cost-to-go), and use another linear layer to predict the action in the next take.

**Models.**  For models written in PDSketch, we have three variants.

1. In PDS-Base, the robot state encodings are treated as a single vector, named `robot-feature`, and the item state encodings are also treated as a single vector (which is the concatenation of their positions and visual features). The action definition encodes no structures. For example,

```
(:action forward
 :parameters (?r - robot)
 :precondition (and )
 :effect (and
   (robot-feature::assign ?r (??f (robot-feature ?r) (item-feature ??)))
 )
)
```

2. In PDS-Abs, we have the abstraction of the idea "robot-facing." This predicate is used to define actions. For example,

```
(:action forward
 :parameters (?r - robot)
 :precondition (and )
 :effect (robot-pose::cond-assign ?r
 (??f
   (robot-pose ?r) (robot-direction-feature ?r)
   (foreach (?o - item)
     (item-feature::cond-select ?o (robot-is-facing ?r ?o))
   )
 )
 (??g (robot-pose ?r) (robot-direction-feature ?r))
 )
)
```

3. In PDS-Rob, the robot's movements have been encoded into the definition. For example,

```
; The first function returns the 2D position that the robot is facing
;   when the robot is at ?p and facing ?d.
;   This function has been implemented externally.
; The second function returns whether an object is an obstacle (so
;   it will block the robot's movement.
;   This information has also been given to the agent as a input.
(:predicates
  (facing [return_type=pose] ?p - pose ?d - direction)
  (is-obstacle ?o - item)
)
(:derived (_is-facing ?p - pose ?d - direction ?t - pose)
  (equal (facing ?p ?d) ?t)
)
(:derived (_robot-facing [return_type=pose] ?r - robot)
  (facing (robot-pose ?r) (robot-direction ?r))
)
(:derived (_robot-is-facing ?r - robot ?o - item)
  (_is-facing (robot-pose ?r) (robot-direction ?r) (item-pose ?o))
)
(:derived (_robot-facing-clear ?r - robot)
  (not (exists (?o - item) (and
    (_robot-is-facing ?r ?o)
    (is-obstacle ?o)
  ))
)
(:action forward
 :parameters (?r - robot)
 :precondition (_robot-facing-clear ?r)
 :effect (and (robot-pose::assign ?r (_robot-facing ?r)) )
)
```

## D.2   Painting Factory

**State space.**   The state space in the painting factory is composed of a list of objects, and a list of containers. Both objects and containers are represented as a tuple of a 3D xyz location, and a image crop. The image crop is generated by first computing the 2D bounding box of the object in the camera plain, cropping out the image patch, and resizing it to 32 by 32. The image encoder is a 3-layer convolutional neural network followed by a linear transformation layer into a 128-dimensional embedding. The xyz location are represented as three numbers, normalized into the range $[0, 1]$.

**Action space.** The action space in the painting factory contain two actions. We first show the PDSketch definition of two actions:

```
(:action move-into
 :parameters (?o - item ?c - container)
 :precondition (and )
 :effect (and
   (item-pose::assign ?o (container-pose ?c))
   (item-feature::cond-assign ?o
     (??g (container-feature ?c))  ; when the container is a bowl,
     (??h (container-feature ?c))  ; paint ?o to be the same color as ?c.
   )
 )
)
(:action move-to
 :parameters (?o - item ?p - pose)
 :precondition (and )
 :effect (and (item-pose::assign ?o ?p))
)
```

The first action, `move-into` is an action defined for each pair of item and container. It changes the pose of the item (now the item will be inside the container). When the item is placed into a bowl, it will be painted into the same color as the bowl. The second action moves the object directly to a designated position.

**Offline data.** Recall that our task is to place two painted objects in the target location to form a designated relationship. To generate demonstrations, we first randomly select two blocks. Next, we paint one of them and place it at the center of the target location. Then, we paint the second object to the designated color and place it with respect to the first object.

**Baselines.** For all baselines, we use a two-layer graph neural network (GNN) to encode the objects. We generate the action in two steps. First, for each (block, container) pair, we generate a scalar logit indicating the score of placing the block into the container. For each block, we also generate a scalar logit indicating the score of directly moving the block. We normalize these logits with a softmax to formulate the action probability. If the chosen action is to directly move the block, we also predict the target location.

**Models.** The key feature in the painting factory domain is that now we need to plan for continuous actions, instead of a discrete space. This is handled using a basic task and motion planning strategy introduced in PDDLStream [Garrett et al., 2020]. We do this in two steps.

First, during training, we build a structured model for classifying geometric relationships (on, left, and right) in the following form:

$$p(a \text{ on } b) = \sigma \left( \frac{\|pose(a) - pose(b) - \theta\|_2^2 - \gamma}{\tau} \right),$$

where $\theta$, $\gamma$, $\tau$ are learnable parameters. $\sigma$ is the Sigmoid function. That is, intuitively, we say the block $a$ is on block $b$ if the pose of $a$ is close to $pose(b) + \theta$. This formulation allows us to "sample" poses of block $a$ given the pose of block $b$.

Thus, during performance time, based on the initial pose of all objects, we sample a set of new poses that are on, left to, and right to the initial poses. We then use these poses to ground concrete operators. This strategy is also called "incremental" sampling in sampling-based task and motion planning literature.