# OpenReview forum: "PDSketch: Integrated Domain Programming, Learning, and Planning"
_NeurIPS.cc/2022/Conference — NeurIPS 2022 Accept_

### Official Review · Reviewer_CyxQ · 2022-07-11

**Rating:** 5
**Confidence:** 2
**Soundness:** 2 fair
**Presentation:** 2 fair
**Contribution:** 2 fair

**Summary:**

The paper proposes PDSketch to generate the object and its features as high-level constructs of the transition model. Compared to earlier tools like PDDL, PDSketch allows vector values in computation and supports learning of unspecified functions from data. PDSketch is a collection of neural networks to convert encode the state, evaluate the goal and build a transition model. The experiments show that PDSketch adapts to novel first order logic goal objectives for navigation in BabyAI and Painting Factory.

**Questions:**

- What is #Obj. Gen. in Table 1?
- PDS-Rob uses predefined transition model, so it seems that it is only learning object properties. Can you clarify how feasible it is obtain the transition model without simulator in real tasks?
- What is the notion of timestep in change of properties? For example, the wetness property changes within the next timestep or based on location change?
- How to compare the capability of PDSketch over existing planning languages in terms of modeling general-purpose TAMP domains?

**Ethics Review Area:**

["I don’t know"]

**Limitations:**

The paper briefly mentions the limitations as not modeling object hierarchy and unable to detect novel objects and properties. Some discussion on potential societal impact can be included as how much modeling time or iterations are needed for human designer to specify the object attributes, or how a system built with PDSketch is easy/hard to update with task specification as compared to more unstructured learning approaches like BC or DT.

**Strengths And Weaknesses:**

The objective of PDSketch is to efficiently model the “local” changes induced by an agent’s actions and “sparse” changes that occur in object properties for generalization in unseen scenarios. The paper describes how predicates, goal evaluation and transition models can be represented as computation graph and learned. Overall, the paper is well-motivated and detailed.

There are assumptions that on object properties that are task relevant, defining its predicates, transition models. This may increase significantly as the different category of objects, their properties and their interdependence increases. Although the authors motivate the use of factored representation of transition model and eval function, this implies that all the objects and their properties are known at test time. The tasks considered are quasi-static, and it is not clear if the proposed approach can be somehow extended to handle dynamics, noisy sensing and actuation of any physical robot. While the proposed approach is shown to perform better than BC or DT and its variants, some stronger baselines can include recent variants of PDDL like [1] and as cited in related work.

[1] Learning symbolic operators for task and motion planning, Tom Silver*, Rohan Chitnis*, Josh Tenenbaum, Leslie Kaelbling, Tomas Lozano-Perez, IROS, 2021

---

> ### Author Response · Authors · 2022-08-02
> **Response to Reviewer CyxQ**
>
> Thank you for your thoughtful reviews.
>
> **Q1**: Assumptions about object properties and transition models.
>
> **A1**: We would like to clarify that users only need to define property names and the structure of the transition model. For example, in Figure 4, we have defined an object property called `color`, but we can leave its recognition model (from images to color properties) as unspecified (the `??` mark). The system will automatically learn from training data, to associate "color embeddings" with each objects (but, in fact, if some other perceptual feature is more useful for making this prediction than the color, then the model would be expected to learn that). That being said, our model, at test time, only assumes knowing how to segment all objects (for extracting features from images), but not their properties (e.g., categories, colors). The learned properties will be inferred from visual inputs using the neural networks associated with the property definitions. Please also refer to our general response.
>
> **Q2**: Dynamics, noisy sensing and actuation.
>
> **A2**: It is completely possible to encode dynamics into the transition model. For example, an pushing action can have pushing force as an additional argument. Meanwhile, it is also possible to make action parameters to be in the robot configuration space (e.g., joint angles) so that they directly translate to robot actuation signals.
>
> However, it is worth mentioning that PDDL-style planning gives us special leverage at the task level, for making long-horizon plans in very large domains by selecting which objects to operate on, in which order.  Although it is possible to encode any discrete-time dynamics in PDDL-sketch rules, it is likely that for a highly dynamic domain that requires control at a high rate, a hierarchical approach would be better, with high-bandwidth closed-loop controllers managing the dynamic interactions and PDSketch encoding reasoning about their preconditions and effects.
>
> Finally, PDSketch can also operate on noisy sensory data, as it already supports neural networks to extract features from raw sensory data. Of course, when sensory error is present or when neural recognition model fails, we need additional modules such as online replanning to recover from failures.
>
> **Q3**: PDDL-Baselines [2].
>
> **A3**: Please see our general response. Paper [2] is not comparable with our framework because it assumes that the predicates are all given to the system, and the predicates are Boolean-valued. By contrast, PDSketch directly learns from images. Second, it also assumes the preconditions and effects are all conjunctive forms. By contrast, PDSketch allows generic functions, such as neural networks. Third, it assumes a known and accurate transition model for planning, the learned abstract model is only for accelerating the search. Our PDSketch framework does not assume this.
>
> [2] Learning symbolic operators for task and motion planning, Tom Silver*, Rohan Chitnis*, Josh Tenenbaum, Leslie Kaelbling, Tomas Lozano-Perez, IROS, 2021
>
> Questions:
>
> **Q4**: What is #Obj. Gen. in Table 1?
>
> **A4**: It refers to the generalization to environments with 6 doors and 8 objects.
>
> **Q5**: PDS-Rob uses predefined transition model, so it seems that it is only learning object properties. Can you clarify how feasible it is to obtain the transition model without simulator in real tasks?
>
> **A5**: Such transition models will be possible to obtain for some robotic motion primitives such as robot base movement, pick and place, and pushing.
>
> **Q6**: The notion of timestep in change of properties.
>
> **A6**: Yes, your interpretation is correct! By convention, for any assignment statement, the properties on the left-hand side are for the next timestep, and the properties on the right-hand side are based on the current step:
>
> ```
> (wetness::assign ?o (??f (wetness ?o)))
> ```
>
> is equivalent to:
>
> ```
> wetness(?o, t+1) = ??f(wetness(?o, t))
> ```
> where t is the current timestep.
>
> **Q7**: Capability compared with existing planning language.
>
> **A7**: PDSketch has the same expressiveness as existing PDDL-based model languages for TAMP (e.g., PDDLStream [3]).
>
> [3] Caelan R. Garrett, Tomás Lozano-Pérez, Leslie P. Kaelbling. PDDLStream: Integrating Symbolic Planners and Blackbox Samplers via Optimistic Adaptive Planning. In ICAPS, 2020.
>
> **Q8**: Limitations.
>
> **A8**: Thank you for the suggestion. We have included new discussions about our limitations in the conclusion section.

---

### Official Review · Reviewer_PhAv · 2022-07-11

**Rating:** 5
**Confidence:** 3
**Soundness:** 2 fair
**Presentation:** 2 fair
**Contribution:** 3 good

**Summary:**

The paper proposes a symbolic language for adding structure to world models in robotics and for defining task-independent heuristics. The language describes predicates, which explain the state of the environment, and actions, which change the state using pre-conditions and effects. Based on the descriptions, neural networks are instantiated to ground the language.


**Questions:**

1. What training data is required by the system? How difficult is it to collect this data on a physical robot?
2. I believe it is assumed that each symbolic action has a pre-defined controller. Could you comment on the difficulties of coming up with the symbolic actions and their controllers in your robotics domain?

**Limitations:**

"Limitations we hope to address include the lack of hierarchy and the inability of the method to discover novel factorizations." – To me, the biggest limitations of this method concern gathering training data with annotated symbols and coming up with a PDSketch description of a domain. I am not saying that these limitations warrant a rejection, but they must be discussed in the paper.

**Strengths And Weaknesses:**

I find it difficult to review this paper because many pages are spent describing the PDSketch language and important model and experiment details are delegated to the appendix. For example, the use of a VQ-VAE for discretization is mentioned in the third-to-last sentence of the methods section. Moreover, the robot experiment, which should be the highlight of this paper, is described in only one paragraph. The scientific contribution might be sufficient for NeurIPS, but the paper needs significant work.

Strengths:
* The paper proposes a general way of automatically adding structure to world models for robotics.
* The proposed method has a fall-back mechanism onto a generic world model, so that the designer does not have to specify every detail of the environment.

Weaknesses:
* There is no discussion of the difficulties of getting training data for the proposed model. It is not clear to me if ground-truth predicates are used to train the predicate classifiers.
* The results show that the model can fit training tasks from few demonstrations, but the evaluation of generalization to novel tasks is limited. Figure 9 shows examples of four goal states, but there are no quantitative results.

# Detailed comments

– Starting on line 34, an argument is made against “classical hand-engineered approaches”. There are no citations and the argument is not revisited in the rest of the paper. I think it would be useful to devote a paragraph to task and motion planning, and how the proposed approach overcomes its limitations.

– [1] is a recent paper that appears to be quite related to this work (e.g. they also put emphasis on sparse transition models). While it is cited in this paper, it is not explicitly discussed. I would like to see a short discussion of the similarities and differences.

– PDS-Abs and PDS-Rob are both described in one sentence. This makes the BabyAI experiment difficult to understand without going into the appendix.

– In the program shown in Figure 5, I think it should be “wetness::assign ?o1 (??h (wetness ?o2))” because we are checking if o2 is a container with paint?

– Line 58: space after comma.

– Reference for [1] has the year 2010 instead of 2019.

– The robotic environment seems to be derived from RAVENS (https://github.com/google-research/ravens) / Transporter Nets. This is not stated explicitly.

– There are a few pre-2018 works that could be cited for graph neural networks. E.g. [2].

# References

[1] Victoria Xia, Zi Wang, and Leslie Pack Kaelbling. Learning Sparse Relational Transition Models. In ICLR, 2019.

[2] M. Gori, G. Monfardini, and F. Scarselli. A new model for learning in graph domains. In Proceedings. 2005 IEEE International Joint Conference on Neural Networks, 2005., volume 2, pages 729–734 vol. 2, 2005.

---

> ### Author Response · Authors · 2022-08-02
> **Response to Reviewer PhAv**
>
> Thank you for your thoughtful reviews.
>
> **Q1**: Paper presentation.
>
> **A1**: Thanks for your suggestion. Given that NeurIPS allows an additional page in camera-ready versions, we have extended the main paper to include additional details. Specifically, we have
>
> - added additional details for VQ-VAE and the implementation of the heurstic generator.
> - added model details for both the BabyAI and the robotic painting environments.
>
> **Q2**: Training data for the model.
>
> **A2**: Please refer to our general response. PDSketch-training only requires trajectories, in the form of a list of (state, action) pairs, the goal formula associated with each trajectory, and a binary signal indicating whether the trajectory successfully achieves the goal. All the feature extractors (e.g., object-image-to-vector encoders), vector-typed predicates (e.g., the color property of objects), the classifiers (e.g., is-red), will be jointly learned, with **NO** additional supervision.
>
> **Q3**: Quantitative evaluation of generalization to novel tasks.
>
> **A3**: Thanks for the suggestion, we have included the results and discussions for model generalization to novel tasks. Specifically, the quantitative performance for these three generalization tasks are: 0.99, 0.98, and 0.87, respectively, measured as success rate after executing the plan. The last task (stacking three objects in a given order) has lower success rate because stacking objects may fail due to controller and physical noises. Future work may consider building closed-loop controller that can recover such failures.
>
> **Q4**: Relationship with Xia et al.
>
> **A4**: Please refer to the general response. The paper from Xia et al. requires that the deictic formula (i.e., the logic formula that refers to relevant objects) can be synthesized with GIVEN symbolic predicates in the input state. By contrast, deictic formula in PDSketch can be represented with neural networks.
>
> **Q5**: Model details for PDS-Abs and PDS-Rob.
>
> **A5**: Thanks for the suggestion. We have updated the paper to include details about these models.
>
> **Q6**: Figure 5 "wetness".
>
> **A6**: In the current version, this branch only updates wetness property of the object based on its current wetness property and the type of box (in the condition of the when statement). For example, when the object is put into a box with water, the object will get wet.
>
> **Q7**: Missing references and typos.
>
> **A7**: Thanks for the suggestion, we have made the changes.
>
> **Q8**: Pre-defined controllers.
>
> **A8**: In a robotic domain, the operators and the controllers are essentially motion primitives: robot-move, pick-up-objects, place-objects, pushing-objects, pouring, stiring etc. Our motivation is that, the controller of these primitives are usually easy to define (for pick-and-place actions, via kinematic solvers) or easy to learn (for simple actions such as pouring and stiring, see [1] for examples). However, it is usually hard to explicitly write down all effects of executing these operators: for example, when pushing a button of a machine, what will be the state change. In this case, PDSketch suggests an learning algorithm that can learn real-world effects of basic robot motion primives.
>
> [1] Learning compositional models of robot skills for task and motion planning. Zi Wang*, Caelan Reed Garrett*, Leslie Pack Kaelbling, Tomás Lozano-Pérez. IJRR, 2021.

---

> > ### Comment · Reviewer_PhAv · 2022-08-05
> > **Response to Rebuttal**
> >
> > Thank you for answering my questions!
> >
> > To check my understanding:
> > * Your model learns to predict the five properties in Figure 3 (pose, color, wetness, dirtiness, type) purely from images without any labels. The reason why it is able to learn these specific variables is that it has detailed information about the preconditions and effects of actions.
> > * VQVAE and decision trees are used to compute heuristic, but they do not model the transition dynamics. Your transition model is fully differentiable.
> >
> > Is this correct?
> >
> > Finally, how do you change the appearance of the objects to indicate that they are wet or dirty?

---

> > > ### Author Response · Authors · 2022-08-06
> > > **Response**
> > >
> > > Thank you for your time and consideration. Your understanding is correct.
> > >
> > > **Q1**: Model learns to predict properties.
> > >
> > > **A1**: Yes. The model learns to predict the properties purely from images. The model can learn because in action definitions you have used these predicates to specify preconditions and effects, and the model learns the grounding of these properties by observing trajectories.
> > >
> > > **Q2**: VQVAE/Decision Tree.
> > >
> > > **A2**: Yes, they are only used for heuristic computation. The transition model is still differentiable.
> > >
> > > **Q3**: Appearance change.
> > >
> > > **A3**: In the simulated environment, we changed the color saturation of objects when they are wet, and we add textures to objects when they are dirty.

---

### Official Review · Reviewer_LBrw · 2022-07-11

**Rating:** 5
**Confidence:** 5
**Soundness:** 3 good
**Presentation:** 4 excellent
**Contribution:** 3 good

**Summary:**

The paper presents a framework for learning environment dynamics models more efficiently by leveraging small amount of human guidance. Specifically, the paper points out that for common robotics tasks, transition models defined in terms of object attributes and relationships are inherently sparse and local (e.g., an action only affects a small subset of the state space), and exploiting such sparsity and locality can improve the data efficiency of model learning. The paper proposes a framework that allows human users to specify such structures with a PDDL-like planning domain language, and the framework converts such specification to a differentiable computational graph that can be optimized using environment interaction data. The method is evaluated on a BabyAI toy environment and a simulated robotic manipulation environment. The paper shows that the resulting sparse model tends to be more sample-efficient and generalizes better to unseen tasks compared to a “dense” object-level transition model.

**Questions:**

Please refer to my main comments ("weaknesses") for questions & comments. They mainly concern limitations of the method and baseline comparison.

**Limitations:**

Like mentioned in the main comment, the method makes assumptions about the primitive skills as well as the nature of environment dynamics. I hope the authors can elaborate on these assumptions and how they can be relaxed in the next revision of the paper.

**Strengths And Weaknesses:**

Overall I enjoyed reading the paper — the idea of leveraging a small amount of human priors to build a sparse transition model for online planning is conceptually simple and is well-executed in the paper. I especially like the manner that human prior is introduced. Specifying the action effect sketch using a PDDL-like language and letting the neural network fill in the low-level details is both elegant and practical.

However, I do have some comments that are mainly focused on the generality of the proposed method, especially when one wishes to apply it to more complex manipulation tasks.
The method assumes a finite number of atomic primitive skills. However, it is unclear to me how this can be applied to more practical settings where skills have parameters, e.g., grasping angles and placing targets. One might argue that the neural network can in addition take in the continuous parameters as input to model the effects of continuously-parameterized skills. But the fact that the skills are not atomic would likely make the locality & sparsity assumption invalid. For example, a grasp may fail, and the end effector may accidentally bump into other objects. In addition, it seems like the paper assumes fully observed & deterministic transition, which is not practical. How would the model learning and the online planning process handle stochastic transition? I hope the authors could address these limitations in the next revisions of the paper.

My second set of comments concern the experiment evaluation, especially the choice of baselines.
- First of all, DT seems to be not the best baseline to compare against, as it tends not to perform well with sparse reward, and it’s a model-free method. How about comparing to more powerful & structured transition models such as PlaNet / Dreamer (Hafner et al., 2019)? They also assume observation-space input and sparse reward, which could make a fair comparison.
- Second, how does PDSketch PDSketch compare to unsupervised methods that try to discover transition sparsity such as Xia et al., 2018? It would be interesting to see whether an unsupervised method can discover similar sparsity patterns as human supervision and how data efficiency of PDSketch scales with the number of training samples.
- Thirdly, PDSketch also shares some similarities with Regression Neural Networks (Xu et al., 2019), which is not mentioned in the related works. RPN shares the same input & output as PDSketch (object-centric encoding input & first-order logic goal). RPN also assumes that humans provide prior guidance on predicate dependencies, which is similar to the idea of sketching action definitions. I would like to see conceptual and possibly empirical comparison between PDSketch and RPN.
- Finally, can a network learn sparsity by itself? How does the effect of human-specified sparse transition rules scale with the amount of data? Table 4 only compares the sample efficiency between PDSketch and model-free method. This calls for an experiment on how PD-Base or similar model-based approaches scales with the amount of training data.

---

> ### Author Response · Authors · 2022-08-02
> **Response to Reviewer LBrw**
>
> Thank you for your thoughtful reviews.
>
> **Q1**: Continuous parameters.
>
> **A1**: You are right that it is possible to have PDSketch operators with continuous parameters as arguments. For example, the move-to operator in the robot painting domain has two arguments `(?x ?pose)`, where the second argument is a continuous parameter, indicating the target location of object `?x`. And furthermore, it is possible to have neural networks take these continuous parameters as their input to make predictions.
>
> **Q2**: Continuous parameters break sparsity and locality assumptions.
>
> **A2**: We argue that locality still exists in the described example. First, when the grasp fails, it only affects objects near the grasping object (but not objects in another room). Second, there still exist conditional sparsity and locality. For example: when the grasp is safe, the locality is entirely preserved. This can be represented in PDSketch as, for objects around the grasping object, when the grasp is unsafe, their pose will also be changed.
> ```
> (foreach (?y - object)
>   (when (and (??around ?y ?x) (??unsafe ?grasp))
>     (pose-assign ?y (?? ...))
>   )
> )
> ```
> Moreover, in practice, usually we only need to model "safe" transitions. Assuming we can detect grasp failure during execution, we can choose to replan from the world state.
>
> **Q3**: Handling stochastic transitions.
>
> **A3**: Please refer to our general response.
>
> **Q4**: Dreamer Algorithm.
>
> **A4**: Thank you for the suggestion. We have included DreamerV2 [1], which is better optimized for image-based environments than DreamerV1 as a baseline for the BabyAI environment. Compared with BC and DT, we see DreamerV2 achieves slightly improved performance for the basic task, but does not show stronger generalization to environments with more objects. We hypothesize this is because Dreamer still learns a fixed policy for execution. By contrast, PDS uses online planning.
>
> It is not straightforward to directly apply Dreamer in the robotic environment, because the action space is object-centric and may have variable length depending on the number of bricks and bowls.
>
> [1] Mastering Atari with Discrete World Models. Danijar Hafner, Timothy Lillicrap, Mohammad Norouzi, Jimmy Ba. In ICLR, 2021.
>
> **Q5**: Learning transition sparsity.
>
> **A5**: Please refer to our general response. Xia et al. is not applicable in domain we are interested in. Specifically, when synthesizing deictic formulas to refer to relevant objects, they rely on given symbolic predicates in states. By contrast, our model assumes only image inputs.
>
> **Q6**: Comparison with Xu et al. 2019. Regression Planning Networks.
>
> **A6**: Thanks for the suggestion. We have added discussions to our paper. There are two key differences. First, Regression Planning Network (RPN) assumes that all the effects and preconditions can be described with a given set of symbolic predicates. By contrast, PDSketch assumes general continuous-valued vector representations for object properties. Second, RPN assumes the controllers for achieving individual symbolic subgoals, while PDSketch supports general parameterized controllers. For example, in the Minigrid domain, RPN assumes controllers for picking up individual objects, but the primitives in PDSketch are only move-foward, turn-left, turn-right, etc. These low-level primitives does not achieves a particular subgoal but only changes object states.
>
> **Q7**: How does the effect of human-specified sparse transition rules scale with the amount of data?
>
> **A7**: Figure 7 (a) and (b) are designed to answer this question. In this case, we plot the loss of transition prediction (y-axis) w.r.t. the number of demonstration episodes (x-axis). Generally, the more structures you program in, the better the data efficiency.

---

> > ### Comment · Reviewer_LBrw · 2022-08-08
> > **Thanks for the response**
> >
> > Thanks for the response. I do not have more questions.

---

### Author Response · Authors · 2022-08-02
**General Response**


We thank all reviewers for their thoughtful comments and suggestions. In our general response, we wish to address a few questions that were raised by multiple reviewers.

**Q1**: What does the user need to specify during training and test.

**A1**: We assume only the following inputs from users:

- a PDSketch definition, which contains a fixed set of parameterized primitive skills (with corresponding low-level controllers), a fixed set of symbols (not necessarily with interpretation, their interpretation will be learned as neural networks)
- a training data set, which contains trajectories (state-action pair sequences) and the goal formula for each training trajectory. The states in the trajectory may have any representation, including images. No additional labels such as object categories or colors are needed.
- an object detection and tracking algorithm which can be used to detect objects from the input images.

Thus, during both training and inference, our model requires NO labels for object categories/colors/other properties. The classifiers defined in PDSketch (e.g., is-red) will be automatically learned, end-to-end, from the paired trajectories and goal formulas.

**Q2**: Handling stochasticity in transitions.

**A2**: In the current formulation, our framework aims at learning a deterministic transition model for the environment, which may be an approximation of the true stochastic dynamics. There are two possible extensions that we can make to handle stochasticity.

- Extending the prediction of effects to be probabilistic. For discrete predicates, this can be done by modeling categorical distributions (e.g., [1]). For continuous-valued predicates, we can leverage generative models such as VAEs [2]. But in both cases, we expect more data needed in order to fit the distributions. It also requires more complex planning algorithms.
- We can also keep the deterministic model and treat it as an abstract model for planning. This approach can be combined with execution-time replanning. For example, when a grasp fails, we replan based on the new state.

[1] Learning Symbolic Models of Stochastic Domains. Hanna M. Pasula, Luke S. Zettlemoyer, Leslie Pack Kaelbling. JAIR 2005.

[2] Auto-Encoding Variational Bayes. Diederik P Kingma, Max Welling. In ICLR, 2014.

**Q3**: Comparison with earlier work on learning operators [3] and learning sparse transition models [4].

**A3**: We would like to point out that [3] requires as input a set of pre-defined symbolic predicates that map a raw input state into a discrete symbolic relational representation. Thus, their operator learning algorithm is completely based on synthesizing logical formulas combining these predicates. By contrast, PDSketch assumes general image-based input representation, and uses general neural networks to represent action effects. Similarly, [4] also requires that the deictic formula (i.e., the logic formula that refers to relevant objects) can be synthesized with GIVEN symbolic predicates in the input state. By contrast, deictic formula in PDSketch can be represented with neural networks (see the PDS-Abs model in the Minigrid domain as an example).

Overall, PDSketch can work directly with sensory inputs (images) while these baselines [3, 4] cannot. Furthermore, PDSketch can model more complex structures (e.g., nested quantifiers) compared to the baseline [3] (only conjunctive logic formula) and [4] (only deictic references).

[3] Learning symbolic operators for task and motion planning. Tom Silver*, Rohan Chitnis*, Josh Tenenbaum, Leslie Kaelbling, Tomas Lozano-Perez. In IROS, 2021.

[4] Victoria Xia, Zi Wang, and Leslie Pack Kaelbling. Learning Sparse Relational Transition Models. In ICLR, 2019.

**To be continued.**

---

> ### Author Response · Authors · 2022-08-02
> **General Response (Continued)**
>
>
> **Q4**: Learning the locality and sparsity structures.
>
> **A4**: It is completely possible for PDSketch to learn locality and sparsity structures. Here we describe two possible directions.
>
> The first direction is to use the "condition" statement in PDSketch. For locality, for example, when moving in a 2D grid, the next location not only depends on the previous location, but also whether there are objects blocking the agent. PDSketch learns this kinds of locality, without additional supervision, too. For example,
> ```
> (:action move-up
>  :parameters (?agent - robot)
>  :precondition (and ...)
>  :effects (and
>    (assign (robot-pos ?agent) (??f
>      (robot-pos ?agent)
>      (forall (?x - object) (when (??facing ?agent ?x) (object-feature ?x)))
>    ))
>  )
> )
> ```
> To break down this definition, the effect changes the robot-pos property of the agent, and its target value depends on the current robot-pos value, and the object-feature of all ?x's such that (??facing ?agent ?x) is true. Note that here, we name this function ??facing only for human readability, this function will be implemented as a neural network and jointly learned.
>
> PDSketch learns ??f and ??facing using gradient decent, in contrast to a more specific greedy-based iterative algorithm as in Xia et al. This gives us the advantage to model not only object-level, but also feature-level sparsity. Moreover, it supports more complex sparsity structures: for example, consider scenerios where the function ??facing also requires the information of other objects to compute.
>
> Next, feature-level sparsity. This can be similarly, encoded using then `when` operator.
>
> ```
> (:action some-action
>  :parameter (?x - object)
>  :precondition (and ...)
>  :effects (
>    (when (??cond1 ) (assign (feat1 ?x) (...)))
>    (when (??cond2 ) (assign (feat2 ?x) (...)))
>    (when (??cond3 ) (assign (feat3 ?x) (...)))
>  )
> )
> ```
> We assume that there are three (possibly unknown) object features: `feat1`, `feat2`, and `feat3`. If we do not know which features this action will change, we can write a few `when` statements, where the conditions `??cond1` etc. will also be jointly learned. Note that, here we assume the conditions are independent of any arguments (i.e., they are constant).
>
> However, we would like to point out that, because these statements are much more general, compared to stating the specific state variable that the action will change, it generally requires more data to learn.
>
> The second direction, which is not discussed in the paper, would be to combine meta-learning with PDSketch. It is an interesting direction to meta-learn PDSketch programs from existing definitions (e.g., existing PDDL files) and synthesize PDSketch files for novel but similar domains.

---

### Public Comment · ~Chenjia_Bai2 · 2023-06-27
**Code**

Hi, authors,

I cannot find the code from https://pdsketch.csail.mit.edu/, would you please give the released code for use?

Thanks a lot

---

> ### Public Comment · Authors · 2023-06-27
> **Thank you for your interest. Code link updated.**
>
> Hi Chenjia,
>
> Thank you for your interest in our work. We have updated the code link on the website. Please don't hesitate to contact us if you have further questions.
>
> Best,
> Authors

---

### Meta-Review · Area_Chair_Qr3P · 2022-09-03

**Recommendation:** Accept
**Confidence:** Less certain

**Metareview:**

This paper present a new domain definition language aimed at defining high-level structures of transition models. The benefit of using this language is the possibility of injecting human priors into the models thus achieving better learning.

The idea of the paper is novel and bold.

The reviewers highlighted different aspects of how the paper could be improved in terms of clarity and structure (potentially move some of the experiments from the appendix), but generally agreed on the value of its contribution.

I encourage the authors to incorporate all the feedback received.

**Award:**

No

---

### Decision · Program_Chairs · 2022-09-14

Accept